# Collaborative Unpaired Multimodal Sensor Fusion for Image Classification

## Abstract

Multimodal learning typically requires expensive paired data for training and assumes all modalities are available at inference. In many sensor-fusion settings, however, data are collected independently by different institutions, come from heterogeneous sensors, and are not aligned at the sample level. We introduce *Unpaired Multimodal Learning (UML)* as the problem of leveraging semantically related but unaligned data across modalities, without requiring explicit pairing or multimodal inference. This setting arises in satellite sensor fusion, where institutions collect data from diverse sensors (optical, multispectral, SAR), but paired acquisitions are rare and data sharing is restricted. We propose a collaborative framework that combines modality-specific projections with a shared backbone, enabling cross-modal knowledge transfer without paired samples. A key element is post-hoc batch normalization calibration, which adapts the shared model to each modality. Our framework also extends naturally to federated training across institutions. Experiments on multiple satellite sensor fusion benchmarks, and additional visual datasets show consistent improvements over unimodal baselines, with particularly strong gains for weaker modalities and in low-data regimes.

## 1 Introduction

Computer vision and sensing tasks often benefit from integrating data from multiple sources that provide complementary information. In remote sensing, for example, optical and radar imagery capture different physical properties of the Earth's surface. While optical sensors offer high spatial resolution, radar can penetrate cloud cover and better characterize surface structure. Similar sensor fusion configurations also arise in medical imaging (*e.g.*, CT–MRI) and robotics (*e.g.*, RGB–depth). Leveraging such modality diversity during training leads to *multimodal learning* and has the potential to provide more robust and accurate models.

Despite this promise, most existing approaches make two strong assumptions: paired sensor samples are available across modalities during training, and all modalities are accessible at inference time. In practice, neither assumption holds. Data are often collected independently by different institutions, coverage is disjoint, sensors are heterogeneous, and privacy constraints limit sharing. As a result, multimodal datasets are frequently *unpaired*, fragmented, and distributed.

Existing strategies often fall into two extremes. On one side, an unimodal training approach which learns models from a single sensor or modality, without requiring any alignment or collaboration (Helber et al., 2019; Sumbul et al., 2021). On the other side, fully paired fusion that improves accuracy, yet requires costly alignment, depends on scarce co-acquisitions, and typically assumes multimodal inputs at inference (Schmitt et al., 2019; Baltrušaitis et al., 2018). A new line of work explores intermediate solutions that reduce the reliance on pairing. Some methods combine a small set of paired samples with a large unpaired corpus to align shared representations (Yacobi et al., 2025). Others address missing modalities during training or inference by designing imputation strategies or learning models that are robust to incomplete inputs (Wu et al., 2024). There are also approaches for unpaired alignment through pseudo-pairs, cycle consistency, or distillation, particularly in medical imaging and multimodal representation learning (Dou et al., 2020;

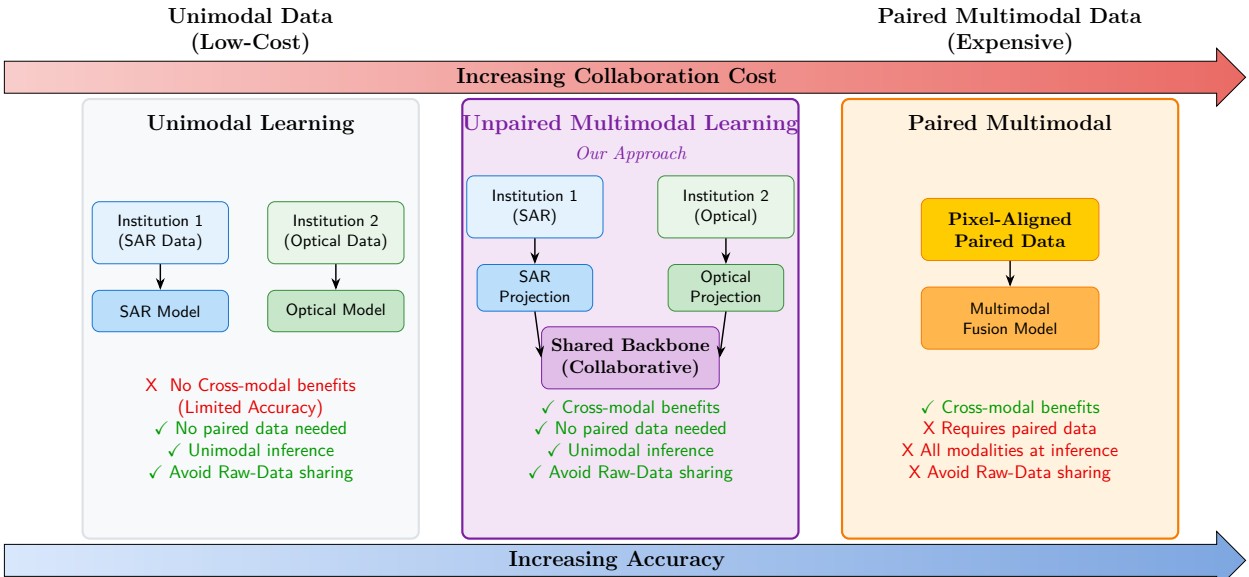

Figure 1: Motivation for unpaired multimodal collaborative learning. At one end, unimodal training is inexpensive and avoid data sharing but fails to exploit complementary information. At the other end, paired multimodal fusion improves accuracy but requires costly aligned data and multimodal inference. Our framework targets the middle ground, enabling cross-modal benefits without paired data or multimodal inputs at test time.

Timilsina et al., 2024). These efforts demonstrate the importance of the unpaired setting, but they typically assume partial pairing, multimodal inference, or homogeneous encoders. In contrast, our goal is to develop a *practical framework for fully unpaired multimodal collaboration* that requires no paired samples, supports unimodal inference, and is simple enough to deploy in realistic distributed settings.

This motivates our central question: *Can we transfer knowledge across modalities without paired data, without multimodal inference, and while respecting heterogeneous architectures and avoiding raw-data sharing?* Figure 1 illustrates this trade-off.

**Motivating example.** Satellite imagery provides a concrete example. Sentinel-1, which carries a *Synthetic Aperture Radar (SAR)* instrument, captures surface structure and is robust to clouds and illumination. Sentinel-2, in contrast, provides multispectral optical imagery with high spatial detail[1]. Each sensor captures heterogeneous modalities, and when paired, they provide shared and unique information. In practice, paired acquisitions across these missions are rare, co-registration is error-prone, and institutions often cannot share raw-data. Similar challenges arise in medical imaging and robotics.

**Our approach.** We propose a framework that enables cross-modal knowledge transfer without requiring paired samples or multimodal inference. Each modality is equipped with a projection into a shared representation, where a backbone learns modality-agnostic semantics. After training, *post-hoc batch normalization calibration* adapts the backbone to each modality, yielding strong unimodal performance.

**Contributions.** This paper makes three contributions:

- **Problem formulation.** We define *Unpaired Multimodal Learning (UML)* as the task of leveraging semantically related but unaligned data across modalities, without requiring paired samples or multimodal inference. Unlike prior settings that assume partial pairing, missing-modality models, or homogeneous encoders, UML captures realistic constraints faced in satellite sensor fusion and related collaborative perception settings.

---

[1]**Sentinel missions.** Sentinel-1 (radar) and Sentinel-2 (optical) are part of the European Copernicus program, which provides freely available global Earth observation data at large scale (Torres et al., 2012; Drusch et al., 2012).

- **Method.** We propose a lightweight collaborative framework for UML that combines modality-specific projections, a shared backbone, and post-hoc batch normalization (BN) calibration (Ioffe & Szegedy, 2015). The design requires no paired data, supports unimodal inference, and we can perform federated training across institutions while not sharing raw-data.

- **Empirical findings.** Across three satellite benchmarks and additional visual datasets, our approach consistently improves over unimodal baselines. Gains are largest for weaker modalities and in low-data regimes, and we show that BN calibration is critical to stable performance. These results establish clear principles for when and why collaboration is most beneficial.

Together, these contributions establish a practical framework for multimodal collaboration under the realistic constraints of unpaired, heterogeneous, and distributed sensor data.

## 2 Problem Formulation

We consider a collaborative learning scenario with $K$ institutions, where each institution $k \in \{1, 2, \ldots, K\}$ holds data from a distinct modality. Let $\mathcal{D}_k = \{(x_i^k, y_i^k)\}_{i=1}^{N_k}$ denote the dataset at institution $k$, where $x_i^k \in \mathcal{X}_k$ represents an input sample from modality $k$, $y_i^k \in \mathcal{Y}$ is the corresponding label, and $N_k$ is the number of samples. Each input space $\mathcal{X}_k$ is heterogeneous, with $\mathcal{X}_k \subset \mathbb{R}^{d_k}$, where the dimensionality $d_k$ may vary across modalities.

**Key constraint (Unpaired Samples).** The datasets are *unpaired* across modalities. This means we do not possess aligned instances $(x^j, x^k)$ that observe the exact same physical entity or location. This eliminates the possibility of pixel-wise or sample-wise alignment.

**Objective.** Each institution aims to learn an improved classifier $h_k(x^k; \omega_k) : \mathcal{X}_k \to \mathcal{Y}$ for its own modality, where $h_k$ is a neural network parameterized by $\omega_k$. The goal is to achieve this improvement by collaborating with other institutions, without sharing raw-data or requiring paired samples.

**Key assumptions.** We make two key assumptions that enable effective collaboration in this challenging setting:

**Assumption 1 (Semantic Coherence).** Although data are unpaired, modalities capture semantically related phenomena and share common high-level semantic structures. Formally, samples from different modalities can be mapped to a shared semantic space $\mathcal{S}$ via functions $\psi_k : \mathcal{X}_k \to \mathcal{S}$. Crucially, these semantic mappings are highly non-injective (many-to-one). Because multiple diverse inputs map to the same semantic concept, the sample-level inverse mapping $\psi_k^{-1}$ is undefined. Therefore, one cannot construct a valid sample-to-sample mapping via composition $(\psi_k^{-1} \circ \psi_j)$, preserving the strictly unpaired nature of the datasets. In our satellite imagery context, this assumption holds because both SAR and optical sensors observe the same Earth surface phenomena, unlike more disparate modality combinations (*e.g.*, text-image). Quantifying semantic coherence prior to training remains an open problem and is beyond the scope of this work. The consistent gains observed across the semantically related modality pairs evaluated here (Tables 1, 2) provide indirect empirical evidence that the assumption holds in our settings.

**Assumption 2 (Shared Label Space).** All institutions operate on the same classification task with identical label space $\mathcal{Y}$. This enables knowledge transfer through supervised learning signals without requiring explicit sample correspondences. While we focus on a shared label space to isolate the effects of multimodal collaboration, this assumption can be relaxed in practice by using task-specific classifier heads for heterogeneous taxonomies.

These assumptions are realistic in many collaborative scenarios: (i) Earth observation, where different institutions collect complementary sensor data (SAR, optical, hyperspectral) for the same land cover classification objectives; (ii) medical imaging, where hospitals may specialize in different modalities (CT, X-ray, MRI) for the same diagnostic task (*e.g.*, lung disease classification), but acquiring paired scans from the same patient across all modalities is rare due to cost and patient burden.

---

**Algorithm 1** Centralized Unpaired Multimodal Learning

---

**Require:** epochs $E$, batch size $b$, number of modalities $K$, learning rate $\eta$
**Require:** per-modality datasets $\{\mathcal{D}_k\}_{k=1}^K$ with $|\mathcal{D}_k| = N$ (balanced); hence $M \triangleq N/b$ mini-batches per epoch

 1: **Initialize:** shared backbone $g(\cdot; \theta)$; modality-specific projections $\{f_k(\cdot; \phi_k)\}_{k=1}^K$
 2: **for** $e = 1$ to $E$ **do**                                                      ▷ epoch
 3:     Shuffle each $\mathcal{D}_k$ and form $M$ mini-batches of size $b$
 4:     **for** $m = 1$ to $M$ **do**                                   ▷ mini-batch within epoch
 5:         **for** $k = 1$ to $K$ **do**
 6:             Sample mini-batch $\{(x_i^k, y_i^k)\}_{i=1}^b$ from $\mathcal{D}_k$
 7:             $\mathcal{L}_k = \frac{1}{b}\sum_{i=1}^b \ell\Big(g\big(f_k(x_i^k; \phi_k); \theta\big), y_i^k\Big)$       ▷ calculate mini-batch loss of each modality
 8:         **Total loss (per step):** $\mathcal{L} = \frac{1}{K}\sum_{k=1}^K \mathcal{L}_k$
 9:         **Update:**   $(\theta, \phi_1, \dots, \phi_K) \leftarrow (\theta, \phi_1, \dots, \phi_K) - \eta\,\nabla_{(\theta, \phi_1, \dots, \phi_K)}\mathcal{L}$
10: **Post-training:** Perform Algorithm 2 for BN calibration

---

## 3 Proposed Approach

### 3.1 Model Architecture

Since our data consist of heterogeneous modalities distributed across many institutions, lets say K institutes, we decompose the modality-specific classifier $h_k(x^k; \omega_k)$ into two components: (i) **Modality-specific projection** $f_k(x^k; \phi_k) : \mathcal{X}_k \to \mathcal{Z}$, which maps raw input $x^k$ to a shared latent space $\mathcal{Z}$; and (ii) **Shared backbone** $g_\theta : \mathcal{Z} \to \mathcal{Y}$, which performs classification in the common representation space. The complete model for modality $k$ is:

$$h_k(x; \omega_k) = g_\theta\big(f_k(x; \phi_k)\big) \qquad \text{s.t.} \qquad x \in \mathcal{X}_k, \tag{1}$$

where $\omega_k = \{\phi_k, \theta\}$ denotes the set of parameters involved for modality $k$.

**Design rationale.** The projection $f_k$ handles modality-specific characteristics (*e.g.*, different channel dimensions, sensor properties), while the shared backbone $g$ learns modality-agnostic semantic features that generalize across modalities.

### 3.2 Training Objective

In the centralized setting, we minimize the empirical risk across all modalities:

$$\mathcal{L}(\phi_1, \dots, \phi_k, \theta) = \frac{1}{K}\sum_{k=1}^K \frac{1}{N_k}\sum_{i=1}^{N_k} \ell\Big(h_k\big(x_i^k\big), y_i^k\Big), \tag{2}$$

and $\ell(\cdot)$ is the cross-entropy loss. Joint optimization encourages $g_\theta$ to learn shared semantics across modalities, akin to multi-task learning where each modality is treated as a related task. Specifically, the shared backbone learns semantically aligned features by observing diverse modality inputs, enabling *implicit cross-modal knowledge transfer*. Weaker modalities benefit from richer representations learned from stronger ones, while stronger modalities gain robustness.

**Training Procedure.** Algorithm 1 presents our training procedure. A critical challenge in multimodal learning is that the batch normalization (BN) statistics computed during training may be biased toward dominant modalities or inappropriate for individual modalities during inference. To address this, we propose a simple yet effective post-training calibration procedure (Algorithm 2). Standard BN uses exponential moving averages to track running statistics during training. However, in the multimodal setting, these statistics represent a mixture across modalities and may not be optimal for any single modality during inference.

---

**Algorithm 2** Post-hoc Batch Normalization (BN) Calibration

---

**Require:** Trained backbone $g(\,\cdot\,;\theta)$, projections $\{f_k(\,\cdot\,;\phi_k)\}_{k=1}^K$
**Require:** Calibration epochs $E_{\text{cal}}$, per-modality datasets $\{\mathcal{D}_k\}_{k=1}^K$, batch size B
  1: **for** $k = 1$ to $K$ **do**
  2:    $g_k(\,\cdot\,;\theta_k) \leftarrow \text{copy}(g(\,\cdot\,;\theta))$ ▷ independent copy
  3:    Freeze all parameters in $g_k$ and $f_k$ ▷ weights fixed
  4:    Reset BN running statistics in $g_k$ and $f_k$ to zero
  5:    Set BN layers to accumulate statistics without momentum (with CMA)
  6:    **for** $e = 1$ to $E_{\text{cal}}$ **do**
  7:       **for** mini-batch $\{x_i^k\}_{i=1}^B \sim \mathcal{D}_k$ **do**
  8:          $Z \leftarrow f_k(\{x_i^k\}_{i=1}^B; \phi_k)$ ▷ batch processing
  9:          $g_k(Z; \theta_k)$ ▷ forward only, updates BN stats
 10:    **Output:** Calibrated model $h_k = g_k \circ f_k$

---

After training, we create modality-specific copies of the shared backbone and recalibrate their BN layers using only training data from the corresponding modality. Specifically, we freeze all learned parameters to preserve trained weights, reset BN running statistics, and disable exponential moving averages. Finally, we recompute statistics using cumulative moving averages (CMA) over multiple calibration epochs ($E_{cal} \ll E$). We use a larger batch size $B \gg b$ during this post-hoc step to minimize sampling noise and improve the accuracy of mean and variance estimates—an approach made possible here because weights are frozen, reducing GPU memory overhead. CMA ensures each batch contributes equally to the final estimate, resulting in more stable statistics than standard exponential moving averages. This procedure requires no additional parameter training and avoids sharing raw-data, as statistics are computed independently for each modality.

**Benefits.** The calibrated models $h_k(\cdot;\omega_k)$ maintain the representational power of the shared backbone while providing modality-appropriate normalization statistics, leading to improved performance without additional learnable parameters or raw-data sharing concerns.

**Connection to multi-task learning.** Our training objective resembles multi-task learning with a shared backbone, where each modality constitutes a distinct "task." This similarity is intentional—it allows our framework to inherit the well-established regularization benefits of multi-task learning, where the shared parameters prevent overfitting to any single modality while learning generalizable representations across tasks.

**Federated Extension.** Our approach naturally extends to federated learning settings (Algorithm 3). The key insight is selective parameter sharing: only the shared backbone parameters $\theta$ are aggregated across clients using FedAvg, while modality-specific projections($\phi_k$ remain local to preserve data heterogeneity and avoid raw-data sharing. This design ensures that raw-data never leaves client devices, with only model parameters being exchanged. Following federated training, each client performs the same BN calibration procedure (Algorithm 2) using local data to obtain personalized models. Our framework is agnostic to the choice of federated aggregation algorithm (FedProx, SCAFFOLD) and optimizer (Adam, SGD), making it broadly applicable to various federated scenarios.

### 3.3 Theoretical Intuition

**Why this works.** The shared backbone $g$ observes diverse feature representations from all modalities during training, acting as a regularizer that prevents overfitting to any single modality's characteristics. Simultaneously, the common classification objective provides a supervisory signal that aligns class-level decision structures across modalities without requiring explicit sample correspondences.

**Cross-modal knowledge transfer.** Weaker modalities benefit from the richer representations learned by stronger modalities through the shared backbone, while stronger modalities gain robustness through exposure to diverse feature patterns.

**BN calibration necessity.** Collaborative training causes Batch Normalization (BN) statistics to drift toward dominant modalities. Post-hoc recalibration using modality-specific data re-centers these distributions, ensuring the global model is correctly mapped to each client's specific feature manifold.

## 4 Experimentation

### 4.1 Datasets and Experimental Setup

We evaluate on three multimodal Earth observation benchmarks: *BigEarthNet-MM* (Sumbul et al., 2021), *EuroSAT-S1-RGB* (Helber et al., 2019; Wang et al., 2024), and *SEN12MS* (Schmitt et al., 2019). All datasets are originally imbalanced and multi-label; we construct class-balanced subsets and recast them as single-label classification to isolate unpaired multimodal learning effects. Each dataset provides Sentinel-1 (S1) SAR and Sentinel-2 (S2) multispectral imagery with varying spectral richness: BigEarthNet-MM (2 SAR + 12 S2 bands), SEN12MS (2 + 13), and EuroSAT-S1/RGB (2 + 3 RGB). The complete data set statistics and band descriptions are in Appendix E and F.

**Training protocol.** We implement our framework with Algorithm 1 primarily in a *centralized* setting. All methods use identical ResNet-18 (He et al., 2016) computational budgets: unimodal baselines train one ResNet-18 per modality, while our method decomposes ResNet-18 into modality-specific projections $f_k$ plus shared backbone $g$ with matching total parameters. Post-training, we recalibrate BatchNorm statistics per modality using local data. We control for all other factors: optimizer, schedule, augmentations, epochs, and early stopping (details in Appendix 13). We report top-1 accuracy (correct predictions/total samples), which is appropriate given the balanced nature of all datasets.

**Experimental configurations.** (i) *Fine-grained*: Each spectral band/channel becomes a separate client ($K = 5$ to $15$), demonstrating scalability; (ii) *Bi-modal*: The standard setting of two clients ($K = 2$), reflecting practical deployment scenarios.

### 4.2 Main Results: Cross-Modal Collaboration Benefits

**Fine-grained collaboration (Experiment 1).** Figure 2a and Figure 5 compare unimodal baselines (blue) to our method (orange) for every band/channel. Our approach improves mean test accuracy on all three datasets: $+\mathbf{12.49}$ percentage points (pp) on BigEarthNet-MM, $+\mathbf{4.59}$ pp on SEN12MS, and $+\mathbf{6.24}$ pp on EuroSAT-S1/RGB.

**Bi-modal collaboration (Experiment 2).** Figure 2b summarizes the bi-modality case. S2 baselines are already high on BigEarthNet-MM and SEN12MS, so adding S1 yields marginal uplifts for S2. In contrast, S1 benefits greatly from collaboration: $+\mathbf{8.4}$ pp (BigEarthNet-MM), $+\mathbf{10.7}$ pp (SEN12MS), and $+\mathbf{7.0}$ pp (EuroSAT-S1-RGB). On EuroSAT, RGB also improves by $+\mathbf{9.5}$ pp. The vertical gray lines denote the performance of a multimodal model trained on paired data. They serve as a reference upper bound, i.e., the performance attainable under conventional multimodal learning with paired supervision. Across three random seeds, our method on BigEarthNet-MM consistently remains below the paired-training baseline, while the differences observed on SEN12MS and EuroSAT-S1 lie within the range of seed-to-seed variability. The only exception is EuroSAT-RGB, where the mean performance is marginally higher than the paired-data reference. However, given the small magnitude of this difference relative to the observed variance across seeds, we do not consider it statistically meaningful and instead attribute it to normal stochastic fluctuations rather than a genuine advantage over paired training. *Takeaway:* collaboration predominantly lifts the weaker modality, while already-informative S2 bands see small but stable changes.

### 4.3 Analysis: Sources of Improvement

**Regularization vs. semantic transfer.** We disentangle two potential mechanisms through controlled experiments (Tables 1, 2 ). Collaborating semantically related modalities (S1 $\leftrightarrow$ S2, MNIST $\leftrightarrow$ SVHN) yields larger gains than unrelated pairs (satellite $\leftrightarrow$ natural images, digits $\leftrightarrow$ fashion). However, even

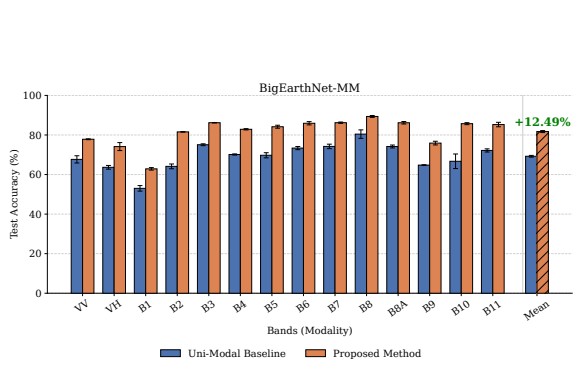

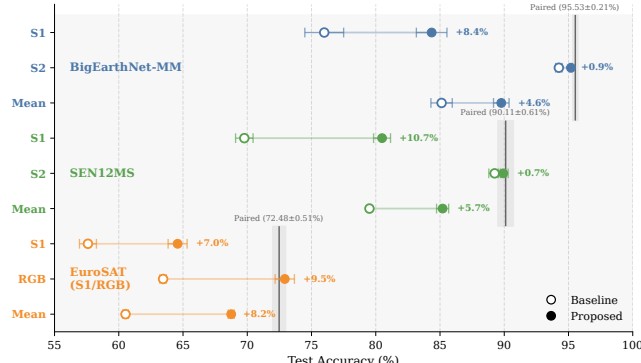

(a) Experiment 1: BigEarthNet-MM (fine-grained)  (b) Experiment 2: Bi-modal (S1 vs. S2/RGB)

Figure 2: **Unpaired collaborative learning consistently improves unimodal classifiers across fine-grained and bi-modal settings.** (Left) Experiment 1: fine-grained per-band accuracy on BigEarthNet-MM comparing unimodal baselines to the proposed joint method; full values in Table 5. (Right) Experiment 2: bi-modal results across three datasets, reported as mean $\pm$ std over 3 seeds. The weaker S1 modality benefits most from bi-modal collaboration. Per-modality values for Experiment 2 are in Table 8. Fine-grained results for SEN12MS and EuroSAT are in Appendix A.

| Method | BigEarthNet-MM | | | SEN12MS | | | EuroSAT S1–RGB | | |
|---|---|---|---|---|---|---|---|---|---|
| | S1 | S2 | Mean | S1 | S2 | Mean | S1 | RGB | Mean |
| Unimodal Baseline | $75.99_{\pm1.51}$ | $94.25_{\pm0.25}$ | $85.12_{\pm0.83}$ | $69.77_{\pm0.68}$ | $89.25_{\pm0.45}$ | $79.51_{\pm0.11}$ | $57.60_{\pm0.66}$ | $63.45_{\pm0.25}$ | $60.52_{\pm0.22}$ |
| ISCA Timilsina et al. (2024) | $82.38_{\pm1.06}$ | $93.48_{\pm0.56}$ | $87.93_{\pm0.54}$ | $72.02_{\pm1.05}$ | $88.04_{\pm0.45}$ | $80.03_{\pm0.74}$ | $\mathbf{65.13}_{\pm1.11}$ | $68.99_{\pm0.90}$ | $67.06_{\pm0.79}$ |
| Proposed (ours) | $\mathbf{84.35}_{\pm1.19}$ | $\mathbf{95.19}_{\pm0.09}$ | $\mathbf{89.77}_{\pm0.60}$ | $\mathbf{80.49}_{\pm0.66}$ | $\mathbf{89.92}_{\pm0.38}$ | $\mathbf{85.20}_{\pm0.47}$ | $64.58_{\pm0.74}$ | $\mathbf{72.92}_{\pm0.75}$ | $\mathbf{68.75}_{\pm0.23}$ |
| Paired (multimodal inference) | – | | $95.53_{\pm0.21}$ | – | | $90.11_{\pm0.61}$ | – | | $72.48_{\pm0.51}$ |

Table 3: Comparison with unpaired multimodal learning methods. Classification accuracy (%) reported as mean $\pm$ std over 3 seeds. **Bold**: best unpaired result per column; Our method outperforms ISCA, the only existing unpaired multimodal approach, and approaches the performance of paired multimodal methods(shown in gray) for reference.

semantically distant collaborators provide positive regularization effects (*e.g.*, SAR+Imagenette[2]: +9.1 pp vs. SAR+optical: +11.7 pp), confirming that both mechanisms contribute.

**Data efficiency and BN calibration.** Figure 3 demonstrates that collaboration benefits are most pronounced in low-data regimes, with diminishing returns as per-modality data increases. Removing BN calibration severely degrades performance across all data scales, with larger degradation at high data volumes—consistent with BN statistics drifting toward dominant modalities during collaborative training.

### 4.4 Baseline Comparisons

**Unpaired multimodal methods.** We compare our approach against Identifiable Shared Component Analysis (ISCA), the only existing method to our knowledge designed for completely unpaired multimodal learning. As shown in Table 3, our method consistently outperforms ISCA across all datasets and modalities.

**Domain adaptation baselines.** Standard DA methods assume shared encoders and often unsupervised targets. We adapt DANN, CDAN, MCC, and MDD to our supervised, heterogeneous-modality setting by using separate encoders per modality with shared classifiers and providing the labels for both modalities(domains). Table 4 shows that our method outperforms all DA variants. Critically, several DA methods underperform unimodal baselines, indicating that standard domain alignment objectives are ill-suited for heterogeneous modalities with different architectural requirements.

---

[2]ImgNette is a subset of Imagenet Dataset containing natural images: `https://github.com/fastai/imagenette`

| Evaluated Modality | Supplementary Modality | | |
|---|---|---|---|
| | MNIST | SVHN | FMNIST |
| MNIST | - | +1.00 | -1.00 |
| SVHN | +3.67 | - | +2.00 |
| FMNIST | -2.00 | +1.67 | - |

Table 1: Cross-modal collaboration benefits on digit datasets. Delta accuracy (%) when adding supplementary modalities. Semantically similar datasets (MNIST+SVHN) show mutual benefits

| Evaluated Modality | Supplementary Modality | | |
|---|---|---|---|
| | BGE-S1 | BGE-S2 | ImgNette |
| BGE-S1 | - | +11.7 | +9.1 |
| BGE-S2 | +1.0 | - | +0.8 |
| ImgNette | +8.4 | +8.7 | - |

Table 2: Cross-modal collaboration on remote sensing data. Delta accuracy (%) with supplementary modalities. BGE-S1/S2 gain most from each other due to semantic similarity.

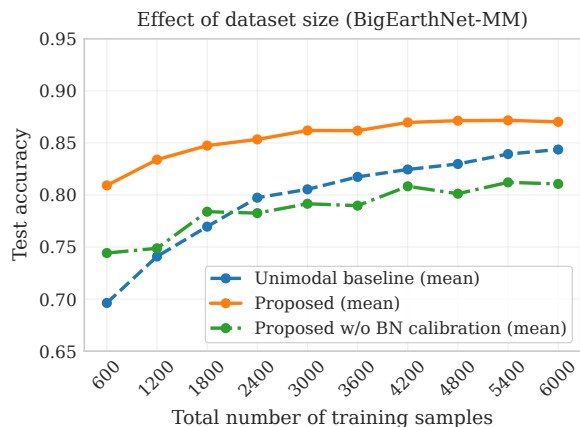

Figure 3: Effect of dataset size on BigEarthNet-MM. Collaboration yields larger gains in the low-data regime, while BN calibration remains critical at all scales.

| Method | BigEarthNet-MM | | | SEN12MS | | | EuroSAT S1–RGB | | |
|---|---|---|---|---|---|---|---|---|---|
| | S1 | S2 | Mean | S1 | S2 | Mean | S1 | RGB | Mean |
| Unimodal Baseline | $75.99_{\pm1.51}$ | $94.25_{\pm0.25}$ | $85.12_{\pm0.83}$ | $69.77_{\pm0.68}$ | $89.25_{\pm0.45}$ | $79.51_{\pm0.11}$ | $57.60_{\pm0.66}$ | $63.45_{\pm0.25}$ | $60.52_{\pm0.22}$ |
| DANN | $70.43_{\pm0.41}$ | $94.24_{\pm0.51}$ | $82.33_{\pm0.45}$ | $69.04_{\pm0.40}$ | $89.06_{\pm0.69}$ | $79.05_{\pm0.29}$ | $58.39_{\pm0.45}$ | $67.72_{\pm1.33}$ | $63.05_{\pm0.67}$ |
| CDAN | $77.85_{\pm0.60}$ | $93.29_{\pm0.84}$ | $85.57_{\pm0.70}$ | $67.13_{\pm2.10}$ | $88.06_{\pm0.99}$ | $77.60_{\pm0.86}$ | $58.33_{\pm0.60}$ | $66.66_{\pm1.00}$ | $62.50_{\pm0.64}$ |
| MCC | $77.40_{\pm0.37}$ | $93.71_{\pm0.83}$ | $85.56_{\pm0.24}$ | $69.82_{\pm2.20}$ | $88.25_{\pm1.76}$ | $79.04_{\pm0.32}$ | $60.63_{\pm0.76}$ | $65.98_{\pm1.59}$ | $63.30_{\pm0.56}$ |
| MDD | $67.53_{\pm1.69}$ | $93.01_{\pm1.00}$ | $80.27_{\pm0.52}$ | $62.06_{\pm2.94}$ | $87.06_{\pm1.30}$ | $74.56_{\pm0.99}$ | $57.21_{\pm2.19}$ | $65.29_{\pm0.47}$ | $61.25_{\pm1.09}$ |
| Proposed (ours) | $\mathbf{84.35}_{\pm1.19}$ | $\mathbf{95.19}_{\pm0.09}$ | $\mathbf{89.77}_{\pm0.60}$ | $\mathbf{80.49}_{\pm0.66}$ | $\mathbf{89.92}_{\pm0.38}$ | $\mathbf{85.20}_{\pm0.47}$ | $\mathbf{64.58}_{\pm0.74}$ | $\mathbf{72.92}_{\pm0.75}$ | $\mathbf{68.75}_{\pm0.23}$ |

Table 4: **Comparison with domain adaptation baselines.** Classification accuracy (%) across three remote sensing datasets. Our method consistently outperforms DA approaches and unimodal baselines. Results demonstrate that our framework achieves superior cross-modal knowledge transfer compared to traditional domain adaptation techniques that require source-target domain alignment.

**Federated Learning Extension.** As discussed earlier, our proposed method naturally extends to federated settings, where no raw-data exchange is required. This is demonstrated in Algorithm 3. Figure 4 shows performance across different local epochs ($L \in \{2, 5, 10, 25, 50\}$). Our method maintains effectiveness across communication-efficiency trade-offs, with optimal performance typically achieved around $L = 5$–$10$ depending on dataset characteristics. Higher local epoch values reduce communication frequency between clients and the server. We denote the number of communication rounds by $R$, where each round involves clients sharing their local backbone weights $\theta_k$ for aggregation. For fair comparison with the centralized setting (200 training epochs), we set $R \in \{100, 40, 20, 8, 4\}$ such that $R \times L = 200$ remains constant across all federated experiments (see Appendix 13 for details).

**Limitations.** Our approach has several important limitations that suggest directions for future work. First, the benefits of collaborative learning diminish as per-modality data becomes abundant (Figure 3). Second, our experimental evaluation focuses on CNN architectures, which are well-suited to the limited-data scenarios that motivate our approach. Modern transformer architectures, like ViTs (Dosovitskiy et al., 2021) with LayerNorm (Ba et al., 2016) or GroupNorm (Wu & He, 2018) may not require BN calibration, as they do not require running statistics. Although ViTs underperform CNNs in the low-data regime, which makes ResNet-18, both a practical and principled choice for the settings studied here. Third, while we evaluate robustness under moderate per-modality size imbalance, up to a 4× ratio between the weaker

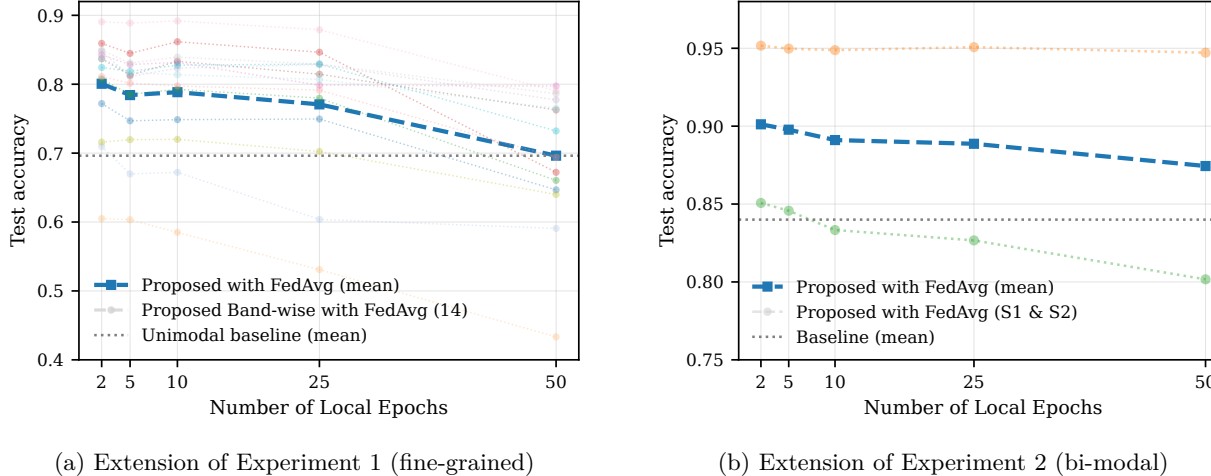

(a) Extension of Experiment 1 (fine-grained)  (b) Extension of Experiment 2 (bi-modal)

Figure 4: Results on BigEarthNet-MM dataset showing the effect of local epochs $L$ on test accuracy under FedAvg with fixed total training budget ($R \times L = 200$). Our method maintains superiority over unimodal baselines across different communication frequencies, with optimal performance at $L = 5$ to $10$. Higher local epochs reduce communication overhead but may degrade performance due to client drift, with convergence to baseline performance at $L = 50$ in both scenarios.

and stronger modality (Table 9), our experiments do not cover severely imbalanced scenarios where one modality contains orders of magnitude more data than others. In such extreme regimes, domination effects during joint training may not be sufficiently mitigated by our current framework, and this remains an open direction. Fourth, our method is designed for the purely unpaired setting and lacks a principled mechanism to leverage partially paired data when available. If some pixel-wise aligned samples exist across modalities, our current framework cannot systematically incorporate this valuable supervisory signal, representing a missed opportunity for improved performance in hybrid scenarios. Fifth, we assume a shared label space and class-balanced, single-label data per modality. Extending the framework to heterogeneous taxonomies, naturally imbalanced data, or multi-label classification remains an important direction for future work. Finally, our federated extension has two open limitations: it assumes balanced label distributions across clients, whereas label skew could bias the shared backbone toward classes favored by dominant clients; and it avoids raw data sharing by exchanging only backbone parameters, but does not provide formal privacy guarantees such as differential privacy. Addressing label skew and incorporating formal privacy mechanisms are both important directions for future work.

## 5 Related Work

**Multimodal representation learning.** Multimodal learning is the process of jointly leveraging information from multiple data modalities (*e.g.*, images, text, audio, or sensor signals) to learn richer and more robust representations than from any single modality alone (Baltrušaitis et al., 2018; Liang et al., 2024). A central challenge is dealing with heterogeneity across modalities and the need for alignment. Traditional fusion methods can be categorized into early, middle, and late fusion. Early fusion concatenates modalities at the input level, middle fusion shares intermediate representations, and late fusion aggregates modality-specific predictions. These approaches often assume paired data and require all modalities during inference. Recent dual-encoder frameworks such as CLIP (Radford et al., 2021), OneLLM (Han et al., 2024), VLMo (Bao et al., 2022), SIMVLM (Wang et al., 2021), ImageBind (Girdhar et al., 2023), VL-GPT (Zhu et al., 2023), and CROMA (Fuller et al., 2023) address modality heterogeneity by assigning each modality its own encoder and aligning representations via contrastive objectives. While effective across heterogeneous modalities, these methods still depend on paired data for training in contrast to our method where we do not need any paired data.

**Domain adaptation.** Our problem of unpaired multimodal learning shares connections with domain adaptation (DA), where the goal is to align feature spaces across source(s) and target(t) domains so that $f(x^{(s)})$ and $f(x^{(t)})$ yield consistent representations when semantically similar (Wilson & Cook, 2020). Popular DA methods include adversarial alignment, *e.g.*, DANN (Ganin et al., 2016) and CDAN (Long et al., 2018), and discrepancy-based approaches such as MDD (Li et al., 2020a) and MCC (Jin et al., 2020). These methods encourage domain-invariant features through a single shared encoder and have proven effective for homogeneous domains. However, they are not directly applicable to multimodal settings, where each modality requires a distinct encoder due to heterogeneous input structures. In our work, we adapt representative DA baselines by equipping each modality with its own encoder. Empirically, these adapted methods struggle to close the modality gap, highlighting their design limitations for unpaired multimodal learning, whereas our approach achieves stronger cross-modal knowledge transfer (see Table 4).

**Multimodal learning with missing or unpaired Data.** Several works extend multimodal learning beyond the fully paired assumption. Nakada et al. (2023) introduce a contrastive framework that integrates unpaired samples into training. Kim & Kim (2024) propose predicting embeddings of missing modalities in the joint representation space to handle incomplete inputs during inference. Ma et al. (2021) address scenarios with severely missing modalities using a meta-learning approach, while large-scale vision-language models such as Singh et al. (2022) leverage self-supervised learning to train on a mix of paired and unpaired data. Timilsina et al. (2024) introduces Identifiable Shared Component Analysis, which disentangles shared and private components from unpaired multimodal distributions to enable a joint classifier. We benchmark against this approach and find that our framework achieves stronger performance while avoiding reliance on paired data (see Table 3)

**Federated learning with heterogeneous data.** Classical FL algorithms such as FedAvg (McMahan et al., 2017), FedProx (Li et al., 2020b), and SCAFFOLD (Karimireddy et al., 2020) assume homogeneous architectures across clients, allowing parameter averaging. Personalization-oriented methods like FedPer (Arivazhagan et al., 2019) and FedRep (Collins et al., 2021) relax this by sharing early layers or representations while keeping task-specific heads local, but they still assume a homogeneous data input structure. More recent approaches, including HeteroFL (Diao et al., 2020) and LG-FedAvg (Liang et al., 2020), support heterogeneous client models or backbone splits, yet they are not designed for fundamentally different modalities. A key challenge in such settings is handling batch normalization (BN) statistics, which become inconsistent across modalities. In our proposed method, running statistics of BN layers are not tracked during training, similar to *static Batch Normalization* (sBN)(Diao et al., 2020). We perform post-hoc BN calibration. This design has two main advantages: (i) it prevents raw-data sharing by avoiding the exchange of first- and second-order statistics across clients, thus addressing one of the key limitations highlighted in Diao et al. (2020); and (ii) it ensures that BN statistics are well aligned with the distributional characteristics of each modality. Notably, this differs from multi-domain batch normalization (BN) methods (e.g., FedBN Li et al. (2021)), which address distribution shifts across domains while operating in a *unimodal* setting. In FedBN, each client maintains its own BN affine parameters and client-specific running statistics. By contrast, we learn a single set of BN affine parameters jointly across all modalities and maintain only modality-specific running statistics. AdaBN (Li et al., 2017), which adapts a model to a new domain by replacing its BN statistics with those computed from target-domain batches. Crucially, AdaBN assumes a single shared input space and feature extractor across domains, whereas our modalities occupy distinct input spaces (e.g., 2-band SAR vs. 12-band multispectral), and one cannot just replace the BN statistics to adapt to a new heterogeneous modality. We perform AdaBN-style calibration on the test set in C, finding a maximum accuracy difference of 0.048 pp across ten data scales and three seeds. Our approach is preferable from a deployment perspective, as it relies solely on the available training data and requires a one-time offline computation prior to deployment, eliminating the need for an additional calibration dataset.

**Multi-Task Learning (MTL).** Hard-parameter-sharing MTL Ruder (2017) is, in some sense, the mirror image of our approach: it shares early layers across tasks and branches into task-specific heads near the output, whereas we share the classifier head and instead use modality-specific projection layers at the input. A related line of work (Bachmann et al., 2022; Bhattacharjee et al., 2023) introduces lightweight modality-specific adapters atop a large, frozen pretrained backbone, restricting training to the adapters and task-

specific heads. In contrast, we assume no pretrained backbone: our shared backbone is trained jointly from scratch across modalities, allowing it to learn modality-invariant semantic representations directly.

## 6 Conclusion and Future Work

We addressed the problem of *Unpaired Multimodal Learning (UML)*, where data from different sensors or modalities are semantically related but not aligned across samples. While prior work has explored partial solutions, such as semi-paired training or missing-modality models, a practical framework for fully unpaired collaboration remained elusive. We proposed a simple yet effective approach that combines modality-specific projections, a shared backbone, and post-hoc BN calibration. This design enables cross-modal knowledge transfer without requiring paired data or multimodal inference, and scales naturally to distributed training across institutions without sharing raw data. Our experiments on three satellite benchmarks, complemented by digits and natural image datasets, demonstrate consistent gains over unimodal baselines, with the strongest improvements for weaker modalities and in low-data regimes. These results show that even lightweight architectural changes can unlock significant cross-modal benefits in realistic unpaired settings. Looking ahead, extensions to transformer backbones, alternative normalization schemes, and richer modality combinations promise to broaden the scope of this framework. This work is a step towards multimodal collaboration under the real-world constraints of unpaired, heterogeneous, and private data.

### Reproducibility statement

We implement all experiments in PyTorch following official reproducibility guidelines[3] with three random seeds. Complete hyperparameters are provided in Appendix 13, and code will be made publicly available upon acceptance.

### Broader Impact Statement

This work introduces a framework for collaborative multimodal learning that eliminates the need for raw-data sharing or sample-level pairing. A primary positive impact of this approach is democratizing machine learning capabilities. It enables institutions with isolated or unimodal datasets to collectively train robust models while strictly maintaining data sovereignty, which is particularly beneficial for fields like Earth observation, healthcare, and scientific research where data silos are prevalent.

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

## Appendix

## A    Additional Experiment 1 Results

In the main text (Section 4.2), we presented Experiment 1 results on BigEarthNet-MM and discussed the general trends across datasets. For completeness, Figure 5 provides the corresponding results for SEN12MS and EuroSAT-S1/RGB. These follow the same pattern: our method consistently improves over mean unimodal baselines.

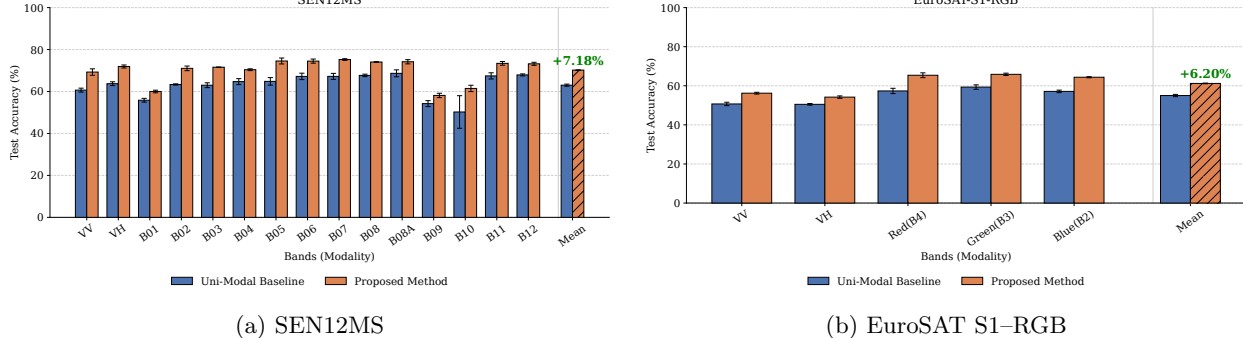

(a) SEN12MS                    (b) EuroSAT S1–RGB

Figure 5: Experiment 1 results for SEN12MS and EuroSAT-S1/RGB. Unimodal baselines (blue) vs. proposed method (orange). Trends are consistent with those reported in the main text. Absolute numbers for accuracy for both datasets are reported in Table 6 and 7

| | Baseline (unimodal) | | Proposed (joint) | | Gain (pp) | |
|---|---|---|---|---|---|---|
| Band | Acc (%) | F1 (%) | Acc (%) | F1 (%) | $\Delta$Acc | $\Delta$F1 |
| VV | 67.67 ± 1.83 | 67.45 ± 1.74 | 77.84 ± 0.22 | 77.84 ± 0.20 | +10.17 | +10.39 |
| VH | 63.66 ± 0.94 | 63.02 ± 0.80 | 74.13 ± 2.02 | 73.65 ± 2.16 | +10.48 | +10.63 |
| B1 | 53.02 ± 1.41 | 52.56 ± 1.16 | 62.88 ± 0.63 | 62.53 ± 0.76 | +9.86 | +9.96 |
| B2 | 64.13 ± 1.20 | 63.85 ± 1.45 | 81.56 ± 0.11 | 81.47 ± 0.13 | +17.43 | +17.63 |
| B3 | 75.07 ± 0.47 | 74.91 ± 0.40 | 86.20 ± 0.12 | 86.18 ± 0.13 | +11.13 | +11.27 |
| B4 | 70.13 ± 0.30 | 69.80 ± 0.46 | 82.87 ± 0.33 | 82.83 ± 0.33 | +12.74 | +13.03 |
| B5 | 69.75 ± 1.29 | 69.52 ± 1.36 | 84.17 ± 0.76 | 84.16 ± 0.75 | +14.42 | +14.65 |
| B6 | 73.38 ± 0.80 | 73.08 ± 0.95 | 85.92 ± 0.83 | 85.92 ± 0.84 | +12.54 | +12.83 |
| B7 | 74.22 ± 1.13 | 73.83 ± 1.09 | 86.21 ± 0.33 | 86.11 ± 0.32 | +11.99 | +12.28 |
| B8 | 80.44 ± 2.14 | 80.25 ± 2.17 | 89.37 ± 0.49 | 89.30 ± 0.46 | +8.93 | +9.05 |
| B8A | 74.18 ± 0.74 | 73.88 ± 0.93 | 86.17 ± 0.63 | 86.09 ± 0.66 | +11.99 | +12.21 |
| B9 | 64.77 ± 0.20 | 63.97 ± 0.36 | 75.90 ± 0.93 | 75.67 ± 0.90 | +11.13 | +11.70 |
| B10 | 66.71 ± 3.69 | 66.49 ± 3.59 | 85.72 ± 0.51 | 85.68 ± 0.47 | +19.02 | +19.20 |
| B11 | 72.19 ± 0.80 | 72.16 ± 0.73 | 85.27 ± 1.13 | 85.26 ± 1.12 | +13.08 | +13.10 |
| Mean | 69.24 ± 0.44 | 68.91 ± 0.43 | 81.73 ± 0.51 | 81.62 ± 0.53 | +12.49 | +12.71 |

Table 5: Absolute accuracy and macro F1 with standard deviation reference for Figure 2a.

## B    Unequal modality dataset sizes

Table 9 evaluates the proposed framework when modalities hold different amounts of training data, a common practical scenario in distributed sensing. We subsample the SAR (S1) modality to 50 and 25 samples per class while the optical modality (S2 or RGB) retains the standard 100 samples per class, yielding 2× and 4× size imbalances, respectively. In the 2× setting (S1: 50, S2: 100), joint training improves macro-F1 for the data-scarce S1 modality by **9.9 pp** on BigEarthNet-MM, **8.1 pp** on SEN12MS, and **8.7 pp** on

| | Baseline (unimodal) | | Proposed (joint) | | Gain (pp) | |
|---|---|---|---|---|---|---|
| Band | Acc (%) | F1 (%) | Acc (%) | F1 (%) | ΔAcc | ΔF1 |
| VV | 60.67 ± 0.93 | 59.82 ± 0.93 | 69.27 ± 1.53 | 68.63 ± 1.58 | +8.61 | +8.81 |
| VH | 63.73 ± 0.90 | 62.46 ± 0.80 | 71.93 ± 0.72 | 71.20 ± 0.78 | +8.20 | +8.74 |
| B01 | 55.83 ± 0.90 | 55.28 ± 1.04 | 60.00 ± 0.63 | 59.76 ± 0.57 | +4.17 | +4.48 |
| B02 | 63.36 ± 0.31 | 63.19 ± 0.36 | 71.04 ± 1.10 | 71.05 ± 1.08 | +7.68 | +7.86 |
| B03 | 63.00 ± 1.15 | 62.65 ± 1.19 | 71.58 ± 0.10 | 71.52 ± 0.16 | +8.58 | +8.87 |
| B04 | 64.71 ± 1.42 | 64.62 ± 1.37 | 70.39 ± 0.43 | 70.37 ± 0.51 | +5.68 | +5.75 |
| B05 | 64.83 ± 1.80 | 64.44 ± 1.78 | 74.55 ± 1.42 | 74.45 ± 1.40 | +9.71 | +10.02 |
| B06 | 67.20 ± 1.53 | 66.52 ± 1.46 | 74.49 ± 0.98 | 74.19 ± 0.93 | +7.29 | +7.67 |
| B07 | 67.21 ± 1.43 | 66.54 ± 1.70 | 75.24 ± 0.40 | 74.93 ± 0.21 | +8.02 | +8.38 |
| B08 | 67.68 ± 0.58 | 66.97 ± 0.84 | 74.06 ± 0.17 | 73.71 ± 0.22 | +6.38 | +6.74 |
| B08A | 68.67 ± 1.65 | 67.88 ± 1.78 | 74.19 ± 1.01 | 73.74 ± 1.22 | +5.52 | +5.86 |
| B09 | 54.23 ± 1.42 | 53.48 ± 1.49 | 58.13 ± 1.03 | 57.56 ± 0.86 | +3.90 | +4.08 |
| B10 | 50.24 ± 7.77 | 49.80 ± 7.93 | 61.46 ± 1.52 | 61.33 ± 1.64 | +11.23 | +11.53 |
| B11 | 67.44 ± 1.48 | 66.64 ± 1.77 | 73.42 ± 0.88 | 72.99 ± 1.05 | +5.98 | +6.35 |
| B12 | 67.91 ± 0.48 | 67.65 ± 0.54 | 73.20 ± 0.72 | 73.03 ± 0.79 | +5.29 | +5.38 |
| Mean | 63.02 ± 0.51 | 62.42 ± 0.54 | 70.20 ± 0.13 | 69.90 ± 0.15 | +7.18 | +7.47 |

Table 6: Absolute accuracy with standard deviation reference for Figure 5a.

| | Baseline (unimodal) | | Proposed (joint) | | Gain (pp) | |
|---|---|---|---|---|---|---|
| Band | Acc (%) | F1 (%) | Acc (%) | F1 (%) | ΔAcc | ΔF1 |
| VV | 50.70 ± 0.88 | 50.10 ± 0.80 | 56.22 ± 0.46 | 56.00 ± 0.42 | +5.52 | +5.91 |
| VH | 50.51 ± 0.42 | 49.38 ± 0.63 | 54.20 ± 0.62 | 53.43 ± 0.56 | +3.69 | +4.05 |
| Red(B4) | 57.36 ± 1.35 | 56.70 ± 1.30 | 65.39 ± 1.23 | 64.95 ± 1.37 | +8.03 | +8.25 |
| Green(B3) | 59.34 ± 1.13 | 58.86 ± 1.10 | 65.86 ± 0.60 | 65.49 ± 0.76 | +6.52 | +6.64 |
| Blue(B2) | 57.13 ± 0.65 | 56.28 ± 0.51 | 64.38 ± 0.28 | 63.76 ± 0.41 | +7.24 | +7.48 |
| Mean | 55.01 ± 0.59 | 54.26 ± 0.46 | 61.21 ± 0.17 | 60.73 ± 0.21 | +6.20 | +6.47 |

Table 7: Absolute accuracy with standard deviation reference for Figure 5b.

EuroSAT over the corresponding unimodal baseline. Under the more severe 4× imbalance (S1: 25, S2: 100), the gains grow to **14.0 pp**, **4.4 pp**, and **8.7 pp** respectively, confirming that joint training provides stronger relative benefit when the weaker modality is more data-starved. The stronger modality (S2/RGB) is largely unaffected (≤0.8 pp change on BigEarthNet-MM and EuroSAT); on SEN12MS S2 we observe a small regression (−2.5 pp at 4× imbalance) that is more than offset by the large S1 gain. These results demonstrate that the proposed framework is robust to realistic per-institution dataset-size imbalances.

## C   Comparison with AdaBN-style calibration

AdaBN (Li et al., 2017) is a closely related technique that adapts a pre-trained model to a target domain by replacing its BN running statistics using target test-domain data. Our post-hoc BN calibration follows the same principle: freeze weights, recompute per-modality BN statistics, but uses only the training split rather than test data. Table 10 compares both strategies across ten data scales (100–1000 samples per class, 3 seeds) on BigEarthNet-MM. The two approaches yield statistically indistinguishable results (max $|\Delta| = 0.048$ pp, mean $|\Delta| = 0.017$ pp), confirming that AdaBN is a viable alternative when test data is available. However, AdaBN's reliance on test data is a practical limitation: mean and variance estimates require a sufficiently large test batch to stabilize, and in an online deployment setting, the model may underperform until enough test samples have been observed. Our approach avoids this dependency entirely — training data is always

| Dataset | Modality | Baseline | Proposed | $\Delta$ |
|---|---|---|---|---|
| BigEarthNet-MM | S1 | $75.99 \pm 1.51$ | $\mathbf{84.35} \pm 1.19$ | +8.4 |
| | S2 | $94.25 \pm 0.25$ | $\mathbf{95.19} \pm 0.09$ | +0.9 |
| | Mean | $85.12 \pm 0.83$ | $\mathbf{89.77} \pm 0.60$ | +4.6 |
| SEN12MS | S1 | $69.77 \pm 0.68$ | $\mathbf{80.49} \pm 0.66$ | +10.7 |
| | S2 | $89.25 \pm 0.45$ | $\mathbf{89.92} \pm 0.38$ | +0.7 |
| | Mean | $79.51 \pm 0.11$ | $\mathbf{85.20} \pm 0.47$ | +5.7 |
| EuroSAT (S1/RGB) | S1 | $57.60 \pm 0.66$ | $\mathbf{64.58} \pm 0.74$ | +7.0 |
| | RGB | $63.45 \pm 0.25$ | $\mathbf{72.92} \pm 0.75$ | +9.5 |
| | Mean | $60.52 \pm 0.22$ | $\mathbf{68.75} \pm 0.23$ | +8.2 |
| BigEarthNet-MM | paired | $95.53 \pm 0.21$ | | – |
| SEN12MS | paired | $90.11 \pm 0.61$ | | – |
| EuroSAT (S1/RGB) | paired | $72.48 \pm 0.51$ | | – |

Table 8: Detailed bi-modal classification results (accuracy %) mean $\pm$ std over 3 seeds. **Bold**: proposed; gray: paired multimodal reference.

| | BigEarthNet-MM | | | SEN12MS | | | EuroSAT S1–RGB | | |
|---|---|---|---|---|---|---|---|---|---|
| Method | S1 | S2 | Mean | S1 | S2 | Mean | S1 | RGB | Mean |
| *S1 subsampled to 50 samples/class; S2/RGB at 100 samples/class (2× imbalance)* | | | | | | | | | |
| Unimodal | $73.33_{\pm 1.09}$ | $94.25_{\pm 0.26}$ | $83.79_{\pm 0.56}$ | $66.09_{\pm 1.38}$ | $89.29_{\pm 0.45}$ | $77.69_{\pm 0.72}$ | $51.45_{\pm 0.51}$ | $62.91_{\pm 0.27}$ | $57.18_{\pm 0.29}$ |
| Joint (ours) | $\mathbf{83.21}_{\pm 1.52}$ | $\mathbf{95.08}_{\pm 0.12}$ | $\mathbf{89.14}_{\pm 0.79}$ | $\mathbf{74.20}_{\pm 0.39}$ | $89.21_{\pm 0.04}$ | $\mathbf{81.70}_{\pm 0.19}$ | $\mathbf{60.12}_{\pm 0.67}$ | $\mathbf{74.47}_{\pm 0.11}$ | $\mathbf{67.29}_{\pm 0.30}$ |
| $\Delta$ Gain | +9.87 | +0.83 | +5.35 | +8.11 | −0.08 | +4.01 | +8.66 | +11.56 | +10.11 |
| *S1 subsampled to 25 samples/class; S2/RGB at 100 samples/class (4× imbalance)* | | | | | | | | | |
| Unimodal | $67.09_{\pm 1.04}$ | $94.25_{\pm 0.26}$ | $80.67_{\pm 0.54}$ | $67.24_{\pm 1.48}$ | $89.29_{\pm 0.45}$ | $78.26_{\pm 0.77}$ | $45.80_{\pm 1.16}$ | $62.91_{\pm 0.27}$ | $54.35_{\pm 0.59}$ |
| Joint (ours) | $\mathbf{81.11}_{\pm 0.94}$ | $\mathbf{94.61}_{\pm 0.26}$ | $\mathbf{87.86}_{\pm 0.48}$ | $\mathbf{71.63}_{\pm 0.87}$ | $86.82_{\pm 0.08}$ | $\mathbf{79.23}_{\pm 0.48}$ | $\mathbf{54.54}_{\pm 1.16}$ | $\mathbf{76.62}_{\pm 0.41}$ | $\mathbf{65.58}_{\pm 0.42}$ |
| $\Delta$ Gain | +14.01 | +0.36 | +7.18 | +4.40 | −2.47 | +0.97 | +8.74 | +13.71 | +11.23 |

Table 9: Robustness under unequal modality dataset sizes. Macro-F1 (%) reported as mean $\pm$ std over 3 seeds. S1 is subsampled while the stronger modality (S2/RGB) retains 100 samples/class in all conditions. **Bold**: best result per column within each regime. $\Delta$ Gain $> 0$: improvement, $< 0$: regression (pp). Gains on the data-scarce S1 grow larger as its data budget shrinks, while S2/RGB performance is largely preserved.

available, typically larger, and its statistics are computed offline in a single controlled pass before deployment.

## D  Federated extension of unpaired multimodal learning

We demonstrate the Federated learning extension of our proposed method in Algorithm 3. We also performed experiments on the BigEarthNet-MM dataset. The results are shown in Figure 4.

## E  Information about Bands

For completeness, we list the spectral bands of Sentinel-1 and Sentinel-2 along with their spatial resolution and typical applications. The information was compiled from Sentinel Hub documentation[4][5] and TorchGeo dataset documentation[6].

---

[4] `https://custom-scripts.sentinel-hub.com/sentinel-2/bands/`

[5] `https://custom-scripts.sentinel-hub.com/custom-scripts/sentinel/sentinel-1/`

[6] `https://torchgeo.readthedocs.io/en/stable/api/datasets.html#bigearthnet`

| Samples/class | **Ours** (BN-train) Acc. (%) | **AdaBN** (BN-test) Acc. (%) | $|\Delta|$ (pp) |
|---|---|---|---|
| 100 | $81.74 \pm 0.72$ | $81.75 \pm 0.73$ | 0.009 |
| 200 | $83.75 \pm 0.53$ | $83.78 \pm 0.52$ | 0.026 |
| 300 | $85.05 \pm 0.30$ | $85.06 \pm 0.28$ | 0.014 |
| 400 | $85.79 \pm 0.42$ | $85.84 \pm 0.45$ | 0.048 |
| 500 | $86.29 \pm 0.08$ | $86.29 \pm 0.03$ | 0.005 |
| 600 | $86.58 \pm 0.35$ | $86.60 \pm 0.37$ | 0.024 |
| 700 | $86.92 \pm 0.13$ | $86.90 \pm 0.12$ | 0.012 |
| 800 | $87.09 \pm 0.20$ | $87.10 \pm 0.20$ | 0.004 |
| 900 | $87.13 \pm 0.17$ | $87.10 \pm 0.18$ | 0.028 |
| 1000 | $87.07 \pm 0.24$ | $87.07 \pm 0.20$ | 0.000 |

Table 10: Effect of calibration data split on BN recalibration. Training-split (Ours) and test-split (AdaBN (Li et al., 2017)) statistics yield indistinguishable accuracy across all data scales (BigEarthNet-MM, 3 seeds).

---

**Algorithm 3** Federated Unpaired Multimodal Learning

---

**Require:** Communication rounds $R$, local epochs $L$, learning rate $\eta$ batch size $b$
**Require:** per-modality datasets $\{\mathcal{D}_k\}_{k=1}^{K}$ with $|\mathcal{D}_k| = N$ (balanced); hence $M \triangleq N/b$ mini-batches per epoch
1: **Initialize:** Shared backbone parameters $g(\,\cdot\,;\theta^0)$, modality-specific projections $\{f_k(\cdot;\phi_k)\}_{k=1}^{K}$
2: **for** round $r = 1$ to $R$ **do**
3:     **Server broadcasts** $g(\cdot;\theta^{r-1})$ to all clients
4:     **for** each client $k$ **in parallel do**
5:         Receive $g(\cdot;\theta^{r-1})$ from server
6:         Initialize local backbone: $g(\cdot;\theta^{k,r}) \leftarrow g(\cdot;\theta^{r-1})$
7:         **for** local epoch $l = 1$ to $L$ **do**
8:             Shuffle each $\mathcal{D}_k$ and form $M$ mini-batches of size $b$
9:             **for** $m = 1$ to $M$ **do**             ▷ mini-batch within epoch
10:                 Compute loss: $\mathcal{L}_k = \frac{1}{b}\sum_{i=1}^{b} \ell\Big(g\big(f_k(x_i^k;\phi_k);\theta^{k,r}\big), y_i^k\Big)$
11:                 Update: $\phi^k, \theta^{k,r} \leftarrow (\phi^k, \theta^{k,r}) - \eta \nabla_{(\theta^{k,r})}\mathcal{L}_k$
12:         Send $\theta^{k,r}$ to server             ▷ Only backbone parameters
13:     **Server aggregates:** $\theta^r \leftarrow \frac{1}{K}\sum_{k=1}^{K}\theta^{k,r}$             ▷ FedAvg
14: **post-training:** Perform **Algorithm 2** for BN calibration

---

# F   Information about Datasets Used

In this section, we provide details of the datasets employed in our experiments. We describe the modalities, selected classes, and experimental splits. While BigEarthNet-MM, SEN12MS, and EuroSAT are co-registered benchmarks (the same geographic acquisition exists in both modalities), we convert them to the unpaired setting as follows. Each modality's training set is constructed independently: patches are shuffled with modality-specific random seeds, and a class-balanced subset is drawn independently per modality. As a result, the S1 and S2 training subsets contain different geographic patches in general. During training, each modality uses an independent data loader with its own random sampler; the co-registration index is never accessed. This protocol intentionally discards the available pairing, making the setting strictly harder than one that exploits spatial correspondence. Table 12 summarizes the dataset statistics, and Figure 6 shows representative examples from the visible spectrum.

**BigEarthNet-MM.** The BigEarthNet-MM dataset consists of co-registered Sentinel-1 (SAR) and Sentinel-2 (multispectral optical) image patches.

| Satellite | Bands | Pixel size (m) | Typical Application |
|---|---|---|---|
| Sentinel-2 | B01 (Aerosol) | 60 | Aerosol detection |
| | B02 (Blue) | 10 | Visible range (RGB) |
| | B03 (Green) | 10 | Visible range (RGB) |
| | B04 (Red) | 10 | Visible range (RGB) |
| | B05 (Red Edge 1) | 20 | Vegetation |
| | B06 (Red Edge 2) | 20 | Vegetation |
| | B07 (Red Edge 3) | 20 | Vegetation |
| | B08 (NIR) | 10 | Shorelines, biomass |
| | B8A (Narrow NIR) | 20 | Vegetation |
| | B09 (Water Vapour) | 60 | Water vapour detection |
| | B10 (Cirrus) | 60 | Cloud detection |
| | B11 (SWIR 1) | 20 | Snow, moisture |
| | B12 (SWIR 2) | 20 | Snow, moisture |
| Sentinel-1 | VV | 10 | Texture, backscatter |
| | VH | 10 | Moisture, vegetation structure |

Table 11: Spectral bands of Sentinel-1 and Sentinel-2 with pixel size and typical applications.

- **Features:** Sentinel-1 provides two polarization bands (VV, VH), while Sentinel-2 provides 12 spectral bands (B01–B12, excluding B10) with spatial resolutions of 10–60 m/pixel. All bands were upsampled to 10 m and cropped into $120 \times 120$ patches.

- **Format:** Each image patch is provided as multiple single-channel GeoTIFFs. Labels are originally multi-class, but we retain only single-label samples.

- **Classes:** We selected six representative classes: *Arable land*, *Broad-leaved forest*, *Coniferous forest*, *Marine waters*, *Pastures*, and *Urban fabric*.

Figure 6(a) shows a visible-spectrum sample, and Table 12 lists the distribution.

**SEN12MS.** SEN12MS is a large-scale dataset of Sentinel-1 and Sentinel-2 patches annotated with MODIS land cover labels. We utilize only single-label International Geosphere-Biosphere Program (IGBP) labels provided by the authors.

- **Features:** Sentinel-1 provides VV and VH backscatter (dB scale), while Sentinel-2 provides 13 spectral bands (B01–B12). Patches are $256 \times 256$.

- **Preprocessing:** The original dataset is seasonally partitioned; we retain only the summer subset.

- **Classes:** We select seven land cover types: *Evergreen broadleaf forest*, *Open shrublands*, *Savannas*, *Grasslands*, *Croplands*, *Urban and built-up*, and *Water bodies*.

A representative sample is shown in Figure 6(b).

**EuroSAT S1–RGB.** The EuroSAT dataset contains Sentinel-2 imagery with 10 target classes. For our experiments, we also include Sentinel-1 SAR patches aligned to the same grid, forming an S1–RGB subset. Original EuroSAT has only RGB images, we obtain the SAR from Wang et al. (2024).

- **Features:** RGB bands (B02, B03, B04) from Sentinel-2 and SAR (VV, VH) from Sentinel-1. Each patch is $64 \times 64$.

- **Classes:** All 10 original classes are retained, including *Annual Crop, Forest, Herbaceous Vegetation, Residential, Sea Lake, Highway, Permanent Crop, Industrial, River, Pasture.*

Figure 6(c) illustrates an RGB example.

| Dataset | Classes | Train | Validation | Test | Resolution |
|---|---|---|---|---|---|
| BigEarthNet-MM | 6 | 1800 (100) | 400 | 700 | $120 \times 120$ |
| SEN12MS | 7 | 300 (100) | 400 | 400 | $256 \times 256$ |
| EuroSAT S1–RGB | 10 | 1000 (100) | 400 | 400 | $64 \times 64$ |

Table 12: Dataset statistics used in our experiments. The numbers in parentheses denote the reduced training samples used in low-data settings (Experiment 1 and Experiment 2).

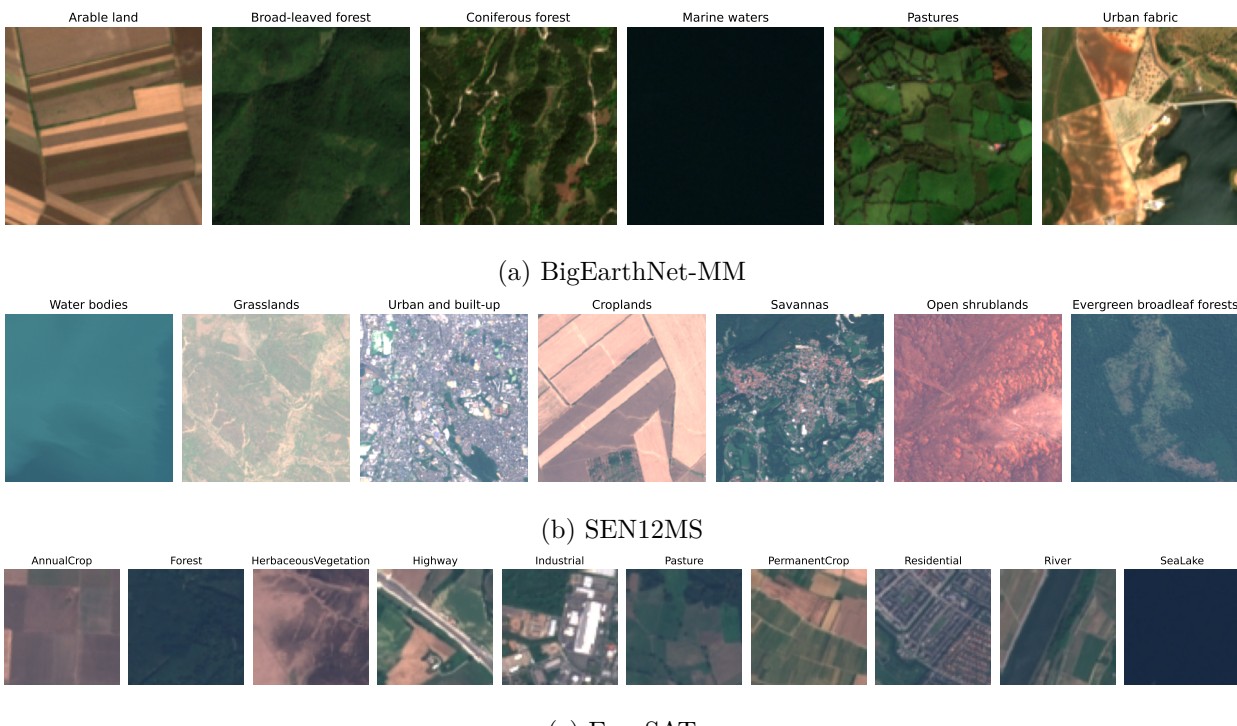

(a) BigEarthNet-MM

(b) SEN12MS

(c) EuroSAT

Figure 6: Representative visible-spectrum examples for selected classes from each dataset.

**ImgNette.** ImgNette is a curated subset of ImageNet designed to reduce label noise and simplify evaluation.

- **Features:** RGB natural images, resized to $224 \times 224$.

- **Classes:** Ten classes corresponding to high-level object categories. For our work, shown in  2, we only used 6 classes.

**MNIST (LeCun, 1998), SVHN (Netzer et al., 2011), and FMNIST (Xiao et al., 2017).** We additionally include canonical vision benchmarks for results shown in  1:

- **MNIST:** Grayscale handwritten digits (10 classes, $28 \times 28$).

- **SVHN:** RGB house numbers trimmed from Google Street View (10 classes, $32 \times 32$). We crop these image to $28 \times 28$ while collaborating with MNIST and Fashion-MNIST.

- **Fashion-MNIST:** Grayscale fashion items (10 classes, $28 \times 28$).

## G   Size of the projection layers vs accuracy

We conducted an ablation to study how the capacity of the modality-specific projection versus the shared backbone affects performance. Specifically, we varied the number of ResNet basic blocks allocated to the projection layer, while keeping the total capacity (projection + backbone) equal to a ResNet-18. As shown in Figure 7, when the projection layer is too deep (and the backbone correspondingly shallow), performance drops significantly (down to ~69%), falling below the unimodal baseline. In contrast, allocating fewer blocks to the projection and more to the backbone yields higher test accuracy (blue curve). This highlights the importance of preserving sufficient depth in the shared backbone.

The orange curve shows results without batch normalization (BN) calibration. The gap clearly demonstrates that BN calibration is critical, especially when the backbone is deeper and contains more BN layers that can drift under multimodal training. Overall, this experiment emphasizes the need for careful capacity allocation between modality-specific and shared components, as well as the necessity of BN calibration.

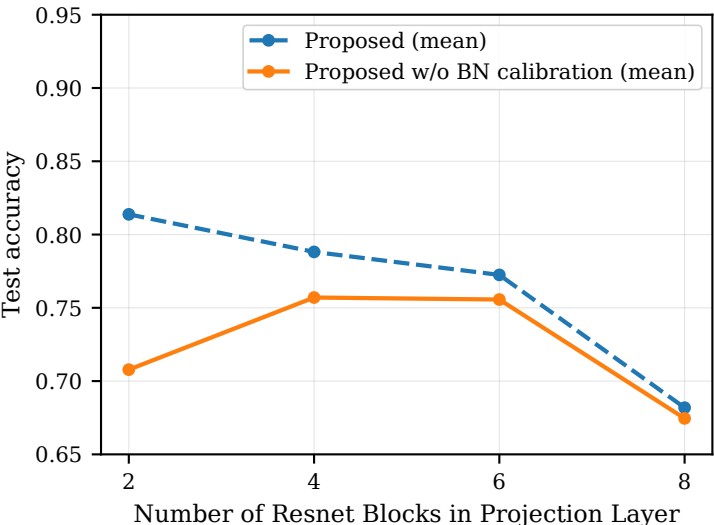

Figure 7: Effect of projection layer depth (in ResNet blocks) on BigEarthNet-MM fine-grained clients (Experiment 1). The blue curve shows results with BN calibration, while the orange curve shows results without.

## H   Use of Large Language Models

We used large language models (ChatGPT-5 and Claude Sonnet 4) as writing assistance tools to improve grammatical correctness and sentence clarity. The LLMs were employed solely for language polishing and did not contribute to research ideation, experimental design, technical methodology, or scientific content generation.

## I   Hyperparameters

The hyperparameters selected for different experiments in our Unpaired multimodal learning are shown in the Table 13

Listing 1: Architecture of the ResNet BasicBlock used in our experiments.

```python
class BasicBlock(nn.Module):
    expansion = 1

    def __init__(self, in_planes, planes, stride=1):
        super(BasicBlock, self).__init__()
        self.conv1 = nn.Conv2d(
            in_planes, planes, kernel_size=3, stride=stride,
            padding=1, bias=False)
        self.bn1 = nn.BatchNorm2d(planes)
        self.conv2 = nn.Conv2d(
            planes, planes, kernel_size=3, stride=1,
            padding=1, bias=False)
        self.bn2 = nn.BatchNorm2d(planes)
        self.shortcut = nn.Sequential()
        if stride != 1 or in_planes != self.expansion * planes:
            self.shortcut = nn.Sequential(
                nn.Conv2d(in_planes, self.expansion * planes,
                          kernel_size=1, stride=stride, bias=False),
                nn.BatchNorm2d(self.expansion * planes)
            )

    def forward(self, x):
        out = F.relu(self.bn1(self.conv1(x)))
        out = self.bn2(self.conv2(out))
        out += self.shortcut(x)
        out = F.relu(out)
        return out
```

Table 13: Hyperparameter selection for different experimental settings on BigEarthNet-MM.

| Hyperparameter | Unimodal | Unpaired Multimodal (Exp. 1 & 2) | Federated UML |
|---|---|---|---|
| Optimizer | AdamW | AdamW | AdamW |
| Weight Decay | 0.01 | 0.01 | 0.01 |
| Initial Learning Rate | 0.001 | 0.001 | 0.001 |
| Batch Size | 32 | 32 | 32 |
| Scheduler | Step | Step | Step |
| Total Epochs | 200 | 200 | 200 |
| Scheduler (step, decay) | (150, 0.1) | (150, 0.1) | (150, 0.1) |
| Augmentations | None (Norm. only) | None (Norm. only) | None (Norm. only) |
| Early Stopping | Best Val ACC after 190 ep. | Best Val ACC after 190 ep. | Best Val ACC after $R{\times}L > 190$ |
| ResNet blocks (projection) | 8 | 2 | 2 |
| ResNet blocks (backbone) | – | 6 | 6 |
| Communication Rounds $R$ | – | – | (100, 40, 20, 8, 4) |
| Local Epochs $L$ | – | – | (2, 5, 10, 25, 50) |
| BN Calibration Epochs | – | 10 | 10 |
| BN Calibration Batch Size | – | 600 | 600 |

