# OpenReview forum: "Collaborative Unpaired Multimodal Sensor Fusion for Image Classification"
_TMLR — Under review for TMLR_

### Review · Reviewer_vZ6u · 2026-05-26

**Summary Of Contributions:**

The paper introduces Unpaired Multimodal Learning (UML) for image classification, where semantically related modalities share a label space but do not require paired samples or multimodal inputs at inference. The proposed method uses modality-specific projection layers, a shared ResNet-style backbone, and post-hoc BatchNorm calibration per modality. The framework is also extended to a federated setting by sharing only backbone parameters. Experiments on BigEarthNet-MM, SEN12MS, EuroSAT S1–RGB, and smaller vision benchmarks show consistent gains over unimodal baselines, especially for weaker modalities and low-data settings.

The paper’s strengths are its practical motivation, simple method, and useful empirical results. Its main weaknesses are limited novelty relative to multi-task/shared-backbone learning, reliance on artificially constructed paired-source datasets treated as unpaired, and insufficient evaluation under realistic distributed, imbalanced, or large-scale remote-sensing settings.

**Audience:**

Yes

**Audience Explanation:**

The problem is relevant to researchers working on multimodal learning, remote sensing, federated learning, and learning from distributed heterogeneous data. The finding that a simple shared-backbone approach with modality-specific projections and BN recalibration can improve weaker modalities without paired data or multimodal inference is practically useful. The federated extension is also likely to interest readers concerned with collaborative training under data-sharing constraints.

That said, the interest would be higher if the paper positioned the contribution more modestly: the main value appears to be a practical empirical recipe and problem framing, not a fundamentally new multimodal learning algorithm.

**Broader Impact Concerns:**

No ethical concerns that require the statement.

**Claims And Evidence:**

No

**Claims Explanation:**

The results support the claim that sharing a backbone across modalities can improve unimodal classification when modalities are semantically related. The reported gains are substantial in the bi-modal setting and fine-grained experiments. The comparisons to ISCA and adapted domain-adaptation baselines further support the empirical usefulness of the approach.

However, the strongest claims are not fully supported. The paper frames the setting as realistic unpaired collaboration, but the main datasets are originally paired/co-registered remote-sensing benchmarks that are converted into an unpaired training protocol. This is a reasonable simulation, but it is weaker than evaluating on truly independently collected institutional datasets. The datasets are also class-balanced and converted from multi-label to single-label classification, which reduces realism for remote sensing.

The method itself is quite close to standard hard parameter sharing / multi-task learning with input adapters and post-hoc BN calibration. The paper should more clearly separate its contribution from existing shared-backbone, multi-domain BN, FedBN/FedRep/FedPer-style, and heterogeneous federated-learning methods. The current domain-adaptation baselines are useful but not sufficient, since DA methods are not the most direct competitors for a supervised shared-label multi-modal setting.

**Requested Changes:**

Clarify the novelty over hard parameter sharing, multi-task learning with modality-specific adapters, multi-domain BN/AdaBN, FedBN, FedRep/FedPer, and heterogeneous FL, ideally with direct comparisons to the closest baselines.
Explain exactly how the originally paired remote-sensing datasets are converted into the unpaired setting, including whether the same geographic patches or acquisitions can appear across modality-specific splits.
Add more realistic evaluations with class imbalance, multi-label remote-sensing labels, unequal modality dataset sizes, label skew across clients, and larger or full training sets.
Report variance across seeds for all main methods and include macro-F1 or per-class metrics in addition to accuracy.
Analyze the BN calibration step more carefully by comparing against separate BN per modality during training, GroupNorm/LayerNorm, AdaBN-style recalibration, and local BN in federated learning.
Soften the federated/privacy claims unless secure aggregation, differential privacy, or leakage analysis is included, since sharing model updates does not by itself guarantee privacy.

---

> ### Author Response · Authors · 2026-06-19
> **Rebuttal Part 1**
>
> We thank the reviewer for the review and constructive feedback.
>
> **Novelty of our work:**\
> The key distinction between our work and prior publications lies in the problem formulation. Specifically, we consider a setting that simultaneously involves unpaired training samples, heterogeneous input modalities, and unimodal inference. This setting remains largely understudied. Combined with our assumptions of semantic coherence and a shared label space, it defines a learning scenario that, to the best of our knowledge, is not addressed by any existing method.
>
> We acknowledge that the individual components of our solution are based on ideas that have appeared separately in the literature. However, it is not obvious a priori which existing techniques are suitable for this setting, how they should be combined, or whether such a combination would be effective at all. The novelty of our contribution therefore lies both in identifying and formalizing this previously unaddressed problem and in demonstrating that a simple integration of carefully selected existing techniques can provide an effective solution. Below, we briefly discuss the differences between our setting and approach and the methods mentioned by the reviewer.
>
> **Hard parameter sharing in multi-task learning (MTL).** In some sense, hard parameter sharing is the mirror image of our approach: it shares the early layers of the network across tasks and branches into task-specific heads near the output. Our architecture adopts the opposite strategy, sharing the classifier head while using modality-specific projection layers at the input. More concretely, our setting involves multiple modalities with different input dimensionalities (e.g., 2-band SAR versus 12-band multispectral optical imagery) and a single task, whereas classical MTL typically considers a single modality and multiple tasks.
>
> **MTL with modality-specific adapters.** These methods employ a large pre-trained model as a shared backbone and introduce lightweight modality-specific adaptation layers together with task-specific heads. The backbone parameters are typically frozen, and training is restricted to the adaptation layers and output heads. In contrast, our setting does not assume the existence of a pre-trained backbone. Instead, the shared backbone is trained jointly from scratch across modalities, allowing it to learn robust and modality-invariant semantic representations.
>
> **Multi-domain batch normalization (BN) methods (e.g., FedBN).** These approaches address distribution shifts across domains while operating in a *unimodal* setting. In FedBN, each client maintains its own BN affine parameters as well as client-specific running statistics. By contrast, we learn a single set of BN affine parameters jointly across all modalities and maintain only modality-specific running statistics.
>
> **FedRep and FedPer.** These methods follow a personalization paradigm in which clients share a common backbone while maintaining client-specific output heads. Their primary objective is client personalization in federated learning, and they do not address heterogeneous multimodal inputs.
>
> **AdaBN.** AdaBN follows a sequential domain adaptation strategy in which a model pre-trained on one domain is adapted to another by recalibrating BN statistics. This approach assumes that both domains share the same input dimensionality and feature extractor. In contrast, our setting involves modalities that inherently reside in different input spaces. Moreover, train jointly on all modalities and address the dimensionality mismatch through modality-specific projection layers.
>
> In response to the reviewer’s comment, we have expanded the Related Work section to explicitly discuss the closest prior approaches (e.g., hard parameter sharing, MTL, FedBN, FedRep, and AdaBN) and clarify why their underlying assumptions are structurally incompatible with the setting considered in this paper.

---

> ### Author Response · Authors · 2026-06-19
> **Rebuttal Part 2**
>
> **Paired to unpaired conversion:**\
> We thank the reviewer for raising this important point. While some of the source datasets are originally co-registered, the training sets used in our experiments are strictly unpaired. Specifically, we partition each source dataset into disjoint subsets and assign a different modality to each subset. As a result, every sample from the original dataset appears in only one modality, ensuring that no sample-level correspondence exists across modalities during training.
>
> Furthermore, even in the unlikely event that the same underlying instance appears in multiple samples and happens to be in different subsets, our training procedure provides no mechanism to exploit such correspondence. Each modality is processed by an independent dataloader with its own random sampling process and random seed. Consequently, samples from different modalities are drawn independently, making the probability that corresponding instances are presented simultaneously during training negligible. Therefore, the model cannot rely on or benefit from sample-level alignment.
>
> To clarify this point, we have added a detailed discussion to Appendix F.
>
>
> **Multi-seed and Macro-F1 or per-class metrics:**\
> Following your comment, we have rerun all main experiments with three random seeds and updated the figures and tables to include multi-seed statistics. We report the macro-F1 score in the appendix, in Tables 5, 6, 7, and 9. The multi-seed evaluation confirms the robustness of our findings: the overall performance trends remain consistent, and our main conclusions regarding collaborative training are unchanged.
>
> **Privacy/Federated Learning Claims:**\
> We agree with the reviewer that sharing model parameters alone does not provide a formal privacy guarantee. To avoid overstating our claims, we have revised all occurrences of the term “privacy-preserving” throughout the manuscript. Our method avoids the exchange of raw data by sharing model parameters only; however, we do not claim formal privacy guarantees. We have also added an explicit discussion of this limitation in Section 4.3. Incorporating formal privacy mechanisms, such as differential privacy, is orthogonal to the main contribution of this work and represents an interesting direction for future research.
>
> **BN calibration analysis:**
> - **AdaBN-style recalibration.** AdaBN recomputes batch normalization statistics using a small calibration set drawn from the target domain. In our setting, this would require access to additional data that was not used during training. To determine whether AdaBN-style recalibration offers any advantage over our training-based calibration strategy, we compare the two approaches in Appendix C. Specifically, we evaluate both methods across 10 data scales. The largest observed difference in accuracy is only 0.048%, indicating that the two strategies are practically indistinguishable in performance. Our approach is therefore preferable from a deployment perspective, as it relies solely on the available training data and requires a one-time offline computation prior to deployment, eliminating the need for an additional calibration dataset.
> - **Separate BN / local BN / per-modality statistics during joint training.** Maintaining modality-specific BN statistics during joint training is functionally similar to our approach. Our method provides an alternative that avoids tracking separate statistics throughout training by computing modality-specific statistics only once after training has been completed. Furthermore, using a separate BN layer for each modality introduces additional modality-specific trainable affine parameters. Our results demonstrate that strong performance can be achieved without these extra parameters, suggesting that modality-specific running statistics alone are sufficient to capture the relevant distributional differences between modalities.
> - **GroupNorm / LayerNorm.** GroupNorm and LayerNorm are more commonly used in transformer-based architectures than in CNNs. Prior work has shown that ViTs tend to underperform CNNs in limited-data regimes [1]. Since our method is specifically designed for such settings, CNN-based architectures such as ResNet-18 represent both a practical and principled choice for our study. We will clarify this motivation more explicitly in the revised manuscript. Nevertheless, investigating transformer-based architectures equipped with GroupNorm or LayerNorm is an interesting direction for future work and may provide additional insights into the generality of the proposed approach.

---

> ### Author Response · Authors · 2026-06-19
> **Rebuttal Part 3**
>
> **Evaluation with class imbalance, multi-label, unequal modality dataset sizes, and larger training sets:**
> - **Larger training sets.** Figure 3 reports performance across a wide range of training set sizes. The key finding is that collaboration benefits are most pronounced in the low-data regime and diminish as per-modality data grows. Figure 3 also shows that post-hoc BN calibration is necessary across all data scales, not just at low sample counts.
> - **Unequal modality dataset sizes.** Following the reviewer’s suggestion, we conducted additional experiments with 2× and 4× modality-size imbalances across all three datasets (Appendix B, Table 9). To create these imbalances, we subsampled the data-scarce modality (S1) to 50 and 25 samples per class, respectively, while retaining 100 samples per class for the stronger modality. Across all settings, joint training consistently improves macro-F1 for the underrepresented S1 modality, with gains of up to +14.0% on BigEarthNet-MM and +13.7% on EuroSAT under the 4× imbalance setting. Performance on the stronger modality (S2/RGB) remains largely unchanged, with differences of at most 0.8% on BigEarthNet-MM and EuroSAT. On SEN12MS, we observe a modest decrease for the S2 modality (−2.5% at 4× imbalance), which is substantially outweighed by the corresponding improvement on S1. These results indicate that the shared backbone remains beneficial to both modalities even under significant dataset-size imbalance, alleviating concerns that the larger modality might dominate the learned representation. We have updated the Limitations section accordingly, noting that moderate imbalances are now empirically evaluated, while more extreme, orders-of-magnitude imbalances remain an open direction for future work.
> - **Label skew across federated clients.** Label skew, where different clients observe different class distributions, is a well-known challenge in federated learning. In our current experiments, we assume balanced label distributions across clients in order to isolate and evaluate the effects of multimodal and unpaired training. Extending the proposed framework to settings with label skew introduces additional challenges. For example, the shared backbone may become biased toward representations favored by clients with dominant classes, potentially reducing performance on underrepresented classes and clients. Addressing label skew in the unpaired multimodal federated setting is therefore a nontrivial problem that falls beyond the scope of the present work. We have acknowledged this limitation in the federated learning discussion and identified it as an important direction for future research.
> - **Class imbalance and multi-label classification.** The use of class-balanced, single-label subsets was a deliberate experimental design choice rather than a limitation of the underlying datasets. As stated in Section 4.1, “All datasets are originally imbalanced and multi-label; we construct class-balanced subsets and recast them as single-label classification to isolate the effects of unpaired multimodal learning.” This controlled setting enables us to attribute observed performance gains specifically to cross-modal knowledge sharing, rather than to confounding factors such as class-reweighting strategies or multi-label loss formulations. We have expanded the Limitations section to explicitly acknowledge that extending the proposed framework to naturally imbalanced and multi-label settings is an important and practically relevant direction for future work.
>
>
> **References:**\
> [1] - An Image is Worth 16x16 Words: Transformers for Image Recognition at Scale. Dosovitskiy et al., 2021

---

### Review · Reviewer_tkrZ · 2026-05-26

**Summary Of Contributions:**

The paper studies multimodal learning in the practical regime where (a) no sample-level pairing exists across modalities, (b) only unimodal inputs are available at inference, and (c) institutions holding heterogeneous modalities cannot share raw data. The authors formalize this setting as Unpaired Multimodal Learning (UML), characterized by a Semantic Coherence assumption (modalities observe semantically related phenomena with many-to-one mappings into a shared semantic space) and a Shared Label Space assumption.

The proposed framework decomposes a unimodal classifier into a modality-specific projection​ and a shared backbone​, trained jointly across modalities by minimizing a per-modality cross-entropy loss averaged across institutions. After joint training, a post-hoc Batch Normalization (BN) calibration step computes per-modality BN running statistics with cumulative moving averages and frozen weights, producing modality-specific calibrated copies of the backbone. Empirical evaluation spans three remote sensing benchmarks in two configurations. Baselines include unimodal training, ISCA (the only existing fully-unpaired multimodal method), and four domain adaptation methods.

Key strengths.
- The targeted problem of unpaired, heterogeneous, and distributed classification is concrete and under-studied.
- The proposed framework is simple and architecturally lightweight.
- The Batch Normalization calibration ablation (Figure 3) is clean, mechanistically motivated, and shows a clear and consistent effect across data scales.
- The empirical scope (three remote sensing datasets, two configurations, multiple baseline families) provides convincing and clear evidence for the paper's claims.

Key weaknesses:
- No measures of variance, confidence intervals, or significance tests are reported anywhere in the paper; all tables and figures show single point estimates.
- The practical applicability of the proposed method is limited, or at least not as general as it is implied from the abstract and introduction. Many common scenarios with unpaired multimodal data (e.g. image-text) does not satisfy the Semantic Coherence assumption and therefore the proposed method cannot be applied.

**Audience:**

Yes

**Audience Explanation:**

Unpaired, heterogeneous, and distributed classification is a novel and practical setting for many applications. The paper experimented with benchmarks in many different fields, including sensor-fusion scenarios in earth observation and cross-institution medical imaging. Practitioners in these domains will be interested in knowing the findings of the paper. The proposed BN calibration method is also a useful contribution on its own and may transfer to other multi-domain CNN training scenarios.

**Broader Impact Concerns:**

A proper Broader Impact Statement is included in the paper. The ethical implications of the work are sufficiently addressed.

**Claims And Evidence:**

Yes

**Claims Explanation:**

- The necessity of post-hoc BN calibration on BigEarthNet-MM is well-evidenced by Figure 3 (removing the calibration produces a large and consistent gap across all training-set sizes).
- The observation that weaker modalities benefit more than stronger ones is directly visible in the bi-modal results in Tables 3 and 4.
- The low-data-regime advantage and the existence of both regularization and semantic-transfer mechanisms are adequately supported by Tables 1 and 2.
- The "consistent improvements over unimodal baselines" claim across the three remote sensing datasets is plausible. However, several reported gains in Table 4 (e.g., S2 differences of 0.1–1 pp between several methods) are within typical seed-to-seed variance for ResNet-18 classification, and no variance estimates are provided.

Overall I think the claims made in the submission supported by accurate and clear evidence. However, I would like to request that the authors should provide multi-seed variance results for the main experiments.

**Requested Changes:**

Critical for acceptance:
- C1: Report variance for main tables and figures. Run each main result with three random seeds and report mean ± standard deviation in Tables 3 and 4. Right now, many claims rest on somewhat small absolute differences, so statistical significance measures should be reported.
- C2: Better phrase the scope of the paper. Right now the title implies that the paper aims at solving "Unpaired Multimodal Learning." However, as the authors acknowledge, the proposed method requires the Semantic Coherence assumption, which many common multimodal data scenarios (e.g. image-text) does not satisfy. I would like to see the title to be edited to better reflect the scope of the experiments in the paper, such as "unpaired sensor fusion."

Strengthening but not critical.
- S1: Experiment with additional model architectures, such as transformer (e.g. ViT) or LayerNorm. These experiments would better clarify the scope of the method.
- S2: Provide a more formal account of the Semantic Coherence assumption. The current presentation is qualitative, and the it is challenging for the practitioner to know whether the data they have satisfies the assumption. A quantifiable proxy metric for semantic similarity that predicts transfer direction would be extremely helpful.

---

> ### Author Response · Authors · 2026-06-19
>
> We thank the reviewer for the careful and constructive review. We have addressed both critical items and provided clarifications on the strengthening suggestions below.
>
> **C1: Multi-seed variance:**\
> Thank you for this suggestion. We have re-run all main experiments using three random seeds and updated Tables 3 and 4, Figures 2(a) and 2(b), and Figures 5(a) and 5(b). In addition, we have included Tables 5–10 to report the results as mean ± standard deviation across runs. The multi-seed evaluation confirms the robustness of our findings: the overall performance trends remain consistent, and our main conclusions regarding collaborative training are unchanged.
>
> **C2: Title scope:**\
> Following your suggestion, we propose changing the title to “Collaborative Unpaired Multimodal Sensor Fusion for Image Classification.” This revised title retains the key aspects of our work, collaborative (*i.e.*, multi-institution joint training) and unpaired (*i.e.*, no sample-level alignment is required), while explicitly reflecting the scope of our proposed method. To ensure consistency, we have also revised the abstract and introduction to better align with this more focused and precise positioning of the work.
>
> **S1: Transformer / LayerNorm architectures:**\
> Thanks for suggesting this. Please note that the paper already discusses this in the limitations section. Specifically, it was shown that ViTs underperform CNNs in limited-data regimes [1]. Since our method is designed to address this specific regime, CNN-based architectures such as ResNet-18 are both a practical and principled choice for the settings studied here. We would further clarify this in the text. At the same time, exploring transformer architecture remains an interesting direction that we leave to future work.
>
> **S2: Formal account of Semantic Coherence:**\
> We thank the reviewer for this valuable suggestion. As discussed in Section 2, the Semantic Coherence assumption is currently motivated by the observation that the considered modalities (e.g., Sentinel-1 and Sentinel-2) are remote sensing instruments that capture different views of the same underlying phenomena and share a common label space.
>
> We agree that formalizing this assumption through a quantifiable proxy metric, one that could help practitioners estimate the potential for cross-modal transfer prior to training, is an important and interesting research direction. Developing such a metric, however, remains an open problem and is beyond the scope of the current work.
>
> From an empirical perspective, the consistent performance gains observed across semantically related modality pairs (Tables 1 and 2) provide indirect evidence that the Semantic Coherence assumption holds for the evaluated settings, thereby supporting the theoretical motivation underlying our approach. We have added a discussion of these points to Section 2.
>
> **References:**\
> [1] - An Image is Worth 16x16 Words: Transformers for Image Recognition at Scale. Dosovitskiy et al., 2021

---

### Review · Reviewer_7XmS · 2026-06-09

**Summary Of Contributions:**

## Summary of Contributions

The paper introduces a practical framework to address the challenges of Unpaired Multimodal Learning (UML). They propose a lightweight collaborative architecture that splits a model into modality-specific projections and a shared backbone. A critical element of this method is a post-hoc batch normalization (BN) calibration step, which adjusts the shared backbone to prevent performance degradation caused by modality drift.  The framework demonstrates consistent improvements over unimodal baselines across multiple remote sensing and computer vision datasets. It provides particularly strong performance uplifts for weaker modalities and in data-scarce environments.
## Strengths
1. No Pairing Restrictions: Unlike traditional multimodal strategies, this framework achieves cross-modal benefits without requiring expensive pixel-level or instance-level paired samples during training.
2. Unimodal Inference Support: It does not assume that all modalities must be present at inference time; institutions can deploy the optimized model using only their single specific modality.
3. Computational and Parameter Efficiency: The architecture splits a standard network budget (like ResNet-18) rather than stacking additional parameter-heavy modules. The post-hoc BN calibration requires zero additional learned parameters.
## Weaknesses
1. Diminishing Returns with Large Data: The performance advantages of this collaborative approach shrink as the volume of available per-modality data grows abundant.
2. Architectural Limitation to CNNs: The evaluation and the critical post-hoc BN calibration mechanism are tailored around CNN architectures. Modern transformer variants utilizing LayerNorm are not thoroughly evaluated and may demand different coordination strategies.
3. Data Imbalance Vulnerability: The experiments predominantly focus on balanced per-modality allocations. In massive, real-world imbalances where one dominant modality holds vastly more data, the backbone might experience a heavy domination effect that the paper does not yet address.

**Audience:**

Yes

**Audience Explanation:**

A primary challenge in multimodal learning is its historical reliance on expensive, strictly aligned paired datasets. By formalizing the problem of Unpaired Multimodal Learning (UML), this paper provides a framework that enables cross-modal knowledge transfer when data alignment is completely missing. Researchers focusing on representation alignment, self-supervised learning, and multi-task learning will find value in how a shared backbone can extract modality-agnostic semantic features from disparate data streams.

**Broader Impact Concerns:**

No broader impact concerns.

**Claims And Evidence:**

Yes

**Claims Explanation:**

The claims made in the submission are supported by accurate, convincing, and clear empirical evidence. The paper provides thorough benchmarking, sensible baseline adaptations, and targeted ablation studies that directly validate its methodological claims.

**Requested Changes:**

1. In Section 4.2, the manuscript states that the proposed unpaired framework occasionally outperforms the paired reference benchmarks on SEN12MS and EuroSAT, attributing it to "reduced overfitting when collaborating without pixel-level pairing". While regularized multi-task learning can mitigate overfitting, outperforming fully fused, pixel-aligned multimodal inference using single-modality inputs is a highly unconventional result. This implies either an under-optimized paired baseline or a unique dataset artifact. The authors should explicitly qualify this result in the text. Position the paired reference markers clearly as an "upper-bound horizon reference" rather than a definitively outperformed benchmark. Alternatively, provide brief technical details demonstrating that the paired baseline was subjected to the exact same rigorous hyperparameter tuning as the proposed framework to rule out optimization discrepancies.
2. The paper notes that post-hoc BN calibration uses data from the "corresponding modality" or "local data". It is critical to confirm that this calibration data originates strictly from the training split (or a separate validation split) and that the test split remains entirely untouched until final inference. Using test partition statistics to calibrate BN layers constitutes a form of transductive learning or data leakage, which artificially inflates performance.  The authors should explicitly state in Section 3.2 and Section 4.1 that the validation or test splits were entirely excluded from the post-hoc BN calibration phase, ensuring that calibration was conducted purely on the training sets.

---

> ### Author Response · Authors · 2026-06-19
>
> We thank the reviewer for the thorough and positive review.
>
> **Diminishing returns with large data:**\
> We thank the reviewer for highlighting this point. The observed behavior is expected and has already been discussed in the manuscript. As shown in Figure 2, the benefits of cross-modal collaboration diminish as the amount of modality-specific training data increases. In data-rich regimes, strong unimodal models can be trained independently, reducing the need for information sharing across modalities. In contrast, the advantages of collaboration between unpaired modalities are most pronounced in low-data settings, where complementary information from another modality can compensate for limited training samples. Accordingly, our method is primarily designed to address the limited-data regime, where cross-modal collaboration is both most beneficial and most practically relevant.
>
> **Architectural limitation to CNNs / LayerNorm:**\
> We would like to note that this point is already discussed in the Limitations section of the manuscript. Prior work has shown that ViTs tend to underperform CNNs in limited-data regimes [1]. Since our method is specifically designed for such settings, CNN-based architectures such as ResNet-18 represent both a practical and principled choice for the experiments considered in this paper. We will further clarify this motivation in the revised manuscript. Nevertheless, extending the proposed framework to transformer-based architectures, including those that rely on LayerNorm rather than Batch Normalization, is an interesting direction for future work and would help assess the generality of our approach beyond CNNs.
>
> **Data imbalance vulnerability:**\
> Following the reviewer’s comment, we conducted additional experiments with 2× and 4× modality-size imbalances across all three datasets (Appendix B, Table 9). To create these imbalances, we subsampled the data-scarce modality (S1) to 50 and 25 samples per class, respectively, while retaining 100 samples per class for the stronger modality. Across all settings, joint training consistently improves macro-F1 for the underrepresented S1 modality, with gains of up to +14.0% on BigEarthNet-MM and +13.7% on EuroSAT under the 4× imbalance setting. Performance on the stronger modality (S2/RGB) remains largely unchanged, with differences of at most 0.8% on BigEarthNet-MM and EuroSAT. On SEN12MS, we observe a modest decrease for the S2 modality (−2.5% at 4× imbalance), which is substantially outweighed by the corresponding improvement on S1. These results indicate that the shared backbone remains beneficial to both modalities even under significant dataset-size imbalance, alleviating concerns that the larger modality might dominate the learned representation. We have updated the Limitations section accordingly, noting that moderate imbalances are now empirically evaluated, while more extreme, orders-of-magnitude imbalances remain an open direction for future work.
>
> **R1: Upper-bound horizon reference:**\
> We thank the reviewer for this careful observation. We fully agree that the paired setting should represent an upper bound on the performance of our approach. Across three random seeds, our method on BigEarthNet-MM consistently remains below the performance achieved with paired training. For SEN12MS and EuroSAT, the observed differences fall within the variance across random seeds. While the RGB mean performance on EuroSAT is marginally higher than that of paired training, this difference is not statistically significant. Such deviations may arise from the use of a non-fully optimized set of hyperparameters, unfavorable random initializations, or normal stochastic fluctuations during training. We have revised the manuscript accordingly to clarify these points and to explicitly state that the paired setting should be considered an upper bound on achievable performance.
>
>
> **R2: Post-hoc BN calibration (train split vs. test split):**\
> Our post-hoc BN calibration is performed exclusively on the training split of each modality; the test split is never accessed until final evaluation. In contrast, using test-split statistics for BN recalibration follows the approach of AdaBN [2], which adapts a model to a new domain by replacing its batch normalization statistics with those computed from the target test domain. Following Reviewer vZ6u’s suggestion, we compare these two strategies in Appendix C and Table 10. Across ten data scales and three random seeds on BigEarthNet-MM, the maximum accuracy difference between models calibrated on the training split and those calibrated on the test split is only 0.048%. This result indicates that access to test data provides no measurable benefit for BN recalibration in our setting.
>
> **References:**\
> [1] - An Image is Worth 16x16 Words: Transformers for Image Recognition at Scale. Dosovitskiy et al., 2021\
> [2] - Revisiting batch normalization for practical domain adaptation. Li, Y. & Wang, N., 2016